



# Using SAR satellite data time-series for regional glacier mapping

Solveig H. Winsvold[1], Andreas Kääb[1], Christopher Nuth[1], Liss M. Andreassen[2], Ward van Pelt[3], and Thomas Schellenberger[1]

[1] Department of Geosciences, University of Oslo, P.O. Box 1047 Blindern, 0316 Oslo, Norway
[2] Section for Glaciers, Ice and Snow, Hydrology Department, Norwegian Water Resources and Energy Directorate, P.O. Box 5091 Majorstua, 0301 Oslo, Norway
[3] Department of Earth Sciences, Uppsala University, Villav. 16, 752 36 Uppsala, Sweden

*Correspondence to*: Solveig H. Winsvold (sohw@nve.no)

## Abstract

With dense SAR satellite data time-series it is possible to map surface and subsurface glacier properties that vary in time. On Sentinel-1A and Radarsat-2 backscatter images over mainland Norway and Svalbard, we have used descriptive methods for outlining the possibilities of using SAR time-series for mapping glaciers. We present five application scenarios, where the first shows potential for tracking transient snow lines with SAR backscatter time-series, and correlates with both optical satellite images (Sentinel-2A and Landsat 8) and equilibrium line altitudes derived from *in situ* surface mass balance data. In the second application scenario, time-series representation of glacier facies corresponding to SAR glacier zones shows potential for a more accurate delineation of the zones and how they change in time. The third application scenario investigates the firn evolution using dense SAR backscatter time-series together with a coupled energy balance and multi-layer firn model. We find strong correlation between backscatter signals with both the modeled firn air-content and modeled wetness in the firn. In the fourth application scenario, we highlight how winter rain events can be detected in SAR time-series, revealing important information about the area extent of internal accumulation. Finally, in the last application scenario, averaged summer SAR images were found to have potential in assisting the process of mapping glaciers outlines, especially in the presence of seasonal snow. Altogether we present examples of how to map glaciers and to further understand glaciological processes using the existing and future massive amount of multi-sensor time-series data. Our results reveal the potential of satellite imagery for automatically derived products as important input in modeling assessments and glacier change analysis.

# 1 Introduction

Glacier change is an important measure of the climate (Vaughan et al., 2013), and glaciers are therefore considered an essential climate variable (GCOS, 2003). Equable baseline datasets are fundamental for parametrization of models and in change analysis (e.g. Fontana et al., 2010; Winsvold et al., 2014; Huss and Hock, 2015), and are essential to better understand the state of glaciers, especially in regions with scarce *in situ* observations.



Optical and synthetic aperture radar (SAR) satellite systems for Earth observation have different advantages and disadvantages for glacier observation. Many of the measured variables included in glacier change studies are often mapped using optical imagery (e.g. Racoviteanu et al., 2009). However, the amount of images is limited in mountain, maritime and high latitude regions due to cloud cover and the polar night. SAR instrument on the other hand, are largely insensitive to weather and, as active instruments, operate independent of solar radiation penetrating clouds. Using SAR and optical imagery in combination should be of high value for understanding processes and for adding information for further advances in glacier remote sensing applications (Kääb et al., 2014).

Glacier mapping is not limited to the detection of glacier outlines, but involves any observation of glacial features and characteristics using remote sensing sources. New methods to map glaciers are needed in view of the higher revisit times of free and open accessible SAR (Sentinel-1A and B) and optical (Landsat 8, Sentinel-2A and B) satellite sensors (Torres et al., 2012; Drusch et al., 2012; Roy et al., 2014). Together, these satellite sensors will enable new multi-sensor time-series applications for mapping glaciers. In this work, we present continuous time-series of Sentinel-1A SAR data (12-day repeat cycle) covering glaciers in Svalbard and mainland Norway. From autumn 2016 two SAR-sensors have been in orbit (Sentinel-1A and B), providing dense temporal sampling availability (6-day repeat).

Backscatter time-series can detect changes in snow and ice conditions, which is related to the amount and variation of ice, air and water in the measured target (e.g. Forster et al., 1996). The received signals at the sensor reflects multiple scattering events dependent on the SAR instrument frequencies, polarization, and imaging geometry, and further the physical characteristics of snow and ice (e.g. roughness), water content, dielectric properties, and volume scattering (Shi and Dozier, 1995; Lillesand et al., 2004, Woodhouse, 2006). Backscatter signals do not only show surface reflectivity, but also signals from below the surface, allowing for differentiating between glacier zones (e.g. Fahnestock et al., 1993; Rau et al., 2000). Studies back to the 1980s have used to investigate snow and ice, and especially using summer SAR images for exploring and tracking transient snow lines (TSL) (Rott, 1984; Bindschadler et al., 1987; Hall et al., 2000; Shi and Dozier, 1994; Rees et al., 1995; Adam et al., 1997; Brown et al., 1999, Casey and Kelly, 2010; Huang et al., 2013; Kundu and Chakraborty, 2015; Callegari et al., 2016). Synergistic use of SAR and optical satellite imagery for mapping snow and ice has been evolving since the 1990s (Rott and Strobl, 1992; Rott, 1994, Shi et al., 1994; Sephton et al., 1995). However, none of these studies use dense continuous Sentinel-1A and B SAR backscatter time-series.

In this paper, we present five application scenarios describing new potential for mapping glaciers with dense SAR satellite image time-series based on robust methods. The chronological order of the imagery, or calculated stack statistics of individual pixels or points (Winsvold et al., 2016), has been analyzed. (1) In the first application scenario, we have tracked the transient snow line (TSL) using SAR time-series data, and we describe the possible connection between TSL's from combined SAR/optical time-series and equilibrium line altitudes (ELA) from *in situ* surface mass balance measurements. (2) Second, we show the stability of winter backscatter values, and a potential of observing glacier facies from 2009-2016 using both Sentinel-1A and Radarsat-2 SAR-data. (3) Third, SAR time-series of surface and subsurface observations have been compared with a surface mass balance and firn evolution model (Van Pelt and Kohler, 2015) using glacier centerlines





profiles. (4) Fourth, we show the potential to map winter rain events over glaciers using high temporal resolution SAR backscatter data. (5) Fifth, we have investigated patterns in summer SAR backscatter signals on- and off-glacier and their potential in glacier outline mapping. For each individual application scenario, a brief introduction, background and method part is given, and results are presented and discussed.

## 2 Study areas

In this paper, we have studied two glaciers near Ny-Ålesund on Svalbard (78.8ºN, 12.7ºE), and five glaciers in southern Norway (61-62ºN, 7-8.6ºE, Fig. 1 and Table 1). Annual mean temperature for Ny-Ålesund is –5.7 ℃, and mean winter and summer temperatures are -12.9 ℃ and 3.7 ℃, respectively (Normal period 1971-2000, Isaksen et al., 2016). Glaciers on Svalbard are maritime and often have a superimposed ice zone (Hagen et al., 2003). Superimposed ice forms from meltwater or rain that refreezes at the surface of the ice (Cogley et al., 2011). Kongsvegen (108 km$^2$) surged in 1948 (Melvold and Hagen, 1998) and is currently in its quiescent phase with low flow velocities (Nuth et al., 2012). Kronebreen (108 km$^2$), the lower part of Holtedahlfonna (295 km$^2$), is a fast-flowing tidewater glacier with maximum ice velocity of 3.2 m d$^{-1}$ at the calving front (Schellenberger et al., 2015).

**Table 1:** Basic glacier information from Nuth and others, 2013 and Andreassen and others, 2012.

| Region | Glacier name | Glacier Ids | | Area (km$^2$) | Aspect | Centerlines | | |
| | | Local | GLIMS | | | Length (km) | Min.elev (m) | Max.elev (m) |
|---|---|---|---|---|---|---|---|---|
| **Svalbard** | Kongsvegen | 15510.1 | G013044E78792N | 108 | North-west | 26 | 0 | 740 |
| | Holtedahlfonna | 15511.2 | G013542E78988N | 295 | South-west | 47 | 0 | 1155 |
| **Norway** | Nigardsbreen | 2297 | G007099E61715N | 42 | South-east | 10.5 | 345 | 1946 |
| | Austdalsbreen | 2478 | G007335E61826N | 10 | South-east | 6 | 1222 | 1755 |
| | Hellstugubreen | 2768 | G008441E61556N | 3 | North-east | 3.4 | 1494 | 2212 |
| | Storbreen | 2636 | G008132E61573N | 5 | North-east | 2.8 | 1398 | 2079 |
| | Gråsubreen | 2743 | G008600E61657N | 2 | North-east | 3 | 1860 | 2399 |

In Southern Norway, the five studied glaciers form a maritime-continental transect and represent diverse glacier characteristics. Nigardsbreen (42 km$^2$) and Austdalsbreen (10 km$^2$) are outlet glaciers from the ice cap Jostedalsbreen. Storbreen (5 km$^2$), Hellstugubreen (3 km$^2$) and Gråsubreen (2 km$^2$) are smaller valley glaciers located in the high mountain area of Jotunheimen. Due to the elevation and more eastward location, these glaciers are more continental than Nigardsbreen and Austdalsbreen. For Norwegian glaciers, the superimposed ice zone is usually absent, even though it may be present on some glaciers (e.g. Brown et al., 2005). The glaciers on mainland Norway all have surface mass balance (SMB) programs measured by the Norwegian Water Resources and Energy Directorate (Kjøllmoen et al., 2016).





## 3 Satellite data

Time-series from two synthetic aperture radar (SAR) sensor types were used in this study, namely Sentinel-1A and B (Interferometric wide swath mode, IW) and Radarsat-2 (Wide, Wide Fine and ScanSAR-Wide modes) (Table 2). Both SAR-sensor types acquire in C-band (center frequency of 5.405 GHz), and images are single polarized SAR data (HH for Svalbard and VV for mainland Norway). Sentinel-1A and B orbits for both Norway and Svalbard are ascending and right looking. Some data gaps exist due to missing images in the download archive (Sentinel-1A) or due to limitations of the data quota and priority (Radarsat-2).

Sentinel-1A and B IW GRD images have a swath of 250 km and a grid-spacing of 10 m, while Radarsat-2 Wide Fine mode has a 150 km swath and a grid-spacing of 8 m. For improved interpretation of a specific 8-day period (in Sect. 5.4) we have also explored Radarsat-2 ScanSAR Wide mode data of 500 km swath with a grid-spacing of 100 m. The images were acquired from different incidence angles and either ascending or descending paths.

Optical satellite data have been included for validation purposes, though with lower temporal resolution mostly due to extended cloud cover in the maritime study regions. Medium resolution optical satellite imagery, Landsat 8 OLI (level-1 Terrain) and Sentinel-2 MSI (level-1C), were both available with radiometrical corrected and ortho-rectified products (Dursch et al., 2012; Roy et al., 2014). These were used for comparison with SAR imagery in the snowline tracking application from Hellstugubreen and Kongsvegen (Sect. 5.1 and Table A5). High mountain areas in Norway are often highly affected by cloud cover, and only seven images were usable for comparison with SAR (three in 2015 and four in 2016). For Kongsvegen on Svalbard, nine optical images were used (five in 2015 and four in 2016). The dates of the optical imagery correlated with the SAR acquisitions by ± 3 days, with two exceptions having a 6-day gap (Table A5). In addition, a Landsat 8 image from 11 September 2015, corresponding to day of year (DOY) 254, was used in the glacier outline mapping (Sect. 5.5).

### 3.1 Additional data

Interpretation of SAR backscatter images for glacier mapping purposes can be challenging since several factors and processes on and within the subsurface affect the phase and magnitude of the SAR signals. The snow, firn and ice are dependent on external and surface conditions, and can result in similar backscatter values. Therefore, additional data, or comparison datasets were used to check the quality and reliability of the SAR time-series. Meteorological data, temperature and precipitation, were downloaded from the Norwegian Meteorological Institute (eKlima.no, 2016) for the Ny-Ålesund meteorological station (Svalbard) and Juvvasshøe meteorological station (mainland Norway) (Fig. A1 and A2). Both stations are located close to the three glaciers used for detailed analysis of the backscatter time-series, namely Hellstugubreen (within ~12 km and station is located 1894 m a.s.l.), Kongsvegen and Holtedahlfonna (within ~20 km and station is located 8 m a.s.l.). We have used existing glacier outlines from both Svalbard (Nuth et al., 2013) and mainland Norway (Andreassen et al., 2008; Paul et al., 2011) as verification and in support of the analysis. Updated glacier outlines for the Norwegian glaciers



are available (Andreassen et al., 2016). The discrepancy between new and old outlines are minor and do not influence the results presented in this paper.

**Table 2:** Overview of Sentinel-1A and B and Radarsat-2 data. * Radarsat-2 images are in Wide (2009-2013) and Wide Fine (2014-2015). The ScanSAR Wide mode is not listed here, as only five images within 8 days was used as ancillary data. ** GRD: Ground range detected images have been detected, multi-looked and projected to ground range coordinates using an ellipsoid (WGS84). *** Sentinel-1B images were available in Norway since 3 October 2016 (DOY 277). Some Sentinel-1A acquisitions were not possible to retrieve from the Sentinels Scientific Data Hub and have caused acquisition gaps in the time-series. For Radarsat-2 the time gaps are due to limitations of the data quota and priority. (Abbreviations: $\theta_i$ = Incident angle, Pol.=Polarization, Temp.res. = Temporal resolution, Rel. path = Relative path, Data prod. = Data product).

| Glacier region | Radarsat-2 * | | | | | | Sentinel-1A and B | | | | | | |
|---|---|---|---|---|---|---|---|---|---|---|---|---|---|
| | # | Pol. | $\theta_i$ | Start date | End date | Temp. res. | # | Pol. | Rel. path | Start date | End date | Data prod ** | Temp. res.*** |
| **Svalbard** | 63 | HH | 37.8° | 8 Feb 2009 | 17 Dec 2015 | 24-days | 37 | HH | 14 | 22 Jan 2015 | 13 Sep 2016 | GRD | 12-days |
| **Norway** | - | - | - | - | - | - | 51 | VV | 44 | 20 Oct 2014 | 15 Oct 2016 | GRD | 12-days (6-days) |

## 4 Processing of SAR imagery

Sentinel-1A and B time-series were processed and geocoded using the open source software SNAP (Sentinel-1 toolbox) distributed by ESA (ESA, 2016), and Radarsat-2 Wide and Wide Fine images were processed and geocoded using the GAMMA remote sensing software (Werner et al., 2000). The Sentinel-1A and B GRD images were in slant range coordinates projected to an ellipsoid, and Radarsat-2 single look complex (SLC) images were in radar geometry. The Sentinel-1 images in SNAP were converted to radiometrically calibrated backscatter in sigma nought ($\sigma^0$) and then filtered using a 3x3 median speckle filter. Furthermore, we applied backscatter terrain correction using a digital elevation model (DEM) and converted linear backscatter values to decibels (dB). The radar backscatter coefficient ($\sigma^0_{dB}$) is hereafter referred to as backscatter. To examine the backscatter results produced in SNAP, we tested the difference between GAMMA and SNAP processed Sentinel-1A GRD images for two winter dates over the glaciers of interest. In GAMMA, the radiometric calibration corrects for two effects: 1) the varying incidence angle on the returned backscatter, and 2) the differing pixel illumination that depends upon the incidence angles (resulting in gamma nought, $\gamma^0$). A speckle filter was not applied on the Sentinel-1 and Radarsat-2 images in GAMMA. However, they were multi-looked using spatial averaging. In north western Svalbard, we used an improved ASTER GDEM (60m) (Nuth et. al., 2013) for geocoding in both software solutions. For scenes covering Norway the SAR images were geocoded using the national 20 m DEM from the Norwegian Mapping Authority (Kartverket, 2016). In Norway and Svalbard, the difference between SNAP and GAMMA processing resulted in a





mean bias of 2.7 dB and 2.4 dB, respectively. We suspect that the bias is a result of the lack of correction in the varying pixel area with incidence angle within the SNAP processing (CCRS, 2002; GAMMA, 2009; step forum, 2016). This correction is normally not needed when the radar backscatter coefficient from one sensor type is analyzed (e.g. Sentinel-1A and B). In addition, the Sentinel-1A scene over north western Svalbard was also processed using a higher resolution IDEM based on

TanDEM-X data. The use of an outdated courser DEM (ASTER GDEM) resulted in very little difference in compared to using a more updated and higher resolution DEM (IDEM from TanDEM-X) over the glacier surfaces. This was as expected given the low slope nature of most glacier surfaces in Svalbard. In addition to the fact that we use repeat paths for the time-series of images, the ASTER GDEM is shown to be more than sufficient for terrain correction. Generally, geometric distortions due to layover, foreshortening and shadowing effects will degrade the SAR images in certain regions, especially

in slopes facing towards the satellite sensor (Woodhouse, 2006).

We assessed combinations of optical and SAR data to analyze image time-series using 1) chronological gap-fill; i.e using SAR data to supplement optical time-series that suffer from heavy cloud cover or missing data due to the dark season. 2) Stack statistics: For a certain time period, SAR pixels are merged by calculating statistics of the time-stack of co-registered pixels (e.g. mean) (Winsvold et al., 2016). A sufficient geocoding and co-registration ensures that the pixels can

be compared in time.

A dense satellite image time-series provides the opportunity to recover a large statistical sample of backscatter values. Heatplots were used for visual inspection of the temporal signature of the glaciers: 1) along the centerline, profile points were selected every 300 m for the Svalbard glaciers (Fig. 2), and 2) the mean of backscatter for elevation zones of 25 m for the study glaciers in mainland Norway (Fig. 4a). Backscattering intensity values are normally very noisy, and the data

from the profiles were smoothed using the mean of 7 x 7 pixel values (30 m pixel size). Representing SAR backscatter data in such a way, we present eleven different mapping variables from the SAR time-series illustrated by numbers on the Kongsvegen profile in Fig. 2.

## 5 Application scenarios

In this section we present application scenarios where we exemplify Sentinel-1 SAR time-series data for glacier mapping

and at the same time discuss the results.

### 5.1 Seasonal melt patterns

The end of summer snowline (EOSS) is an approximation of the equilibrium line altitude (Østrem, 1975; Rabatel et al., 2013). In regions with a significant superimposed ice zone, the EOSS will often be above the ELA (Cogley et al., 2011). In this application scenario, SAR backscatter data from Kongsvegen were used to map glacier surface melt, and to track TSLs

on Hellstugubreen in Norway. Optical satellite data, *in situ* observations and ELA derived from surface mass balance measurements have been used for validation. The EOSS can be used to reconstruct annual mass balance series regionally,



due to the strong relationship between the EOSS and ELA (e.g. Demuth and Pietroniro, 1999; Pelto, 2001). EOSS and TSLs have been used to determine glacier mass balance using modeling assessments (e.g. Rabatel et al., 2005; Huss et al., 2013; Hulth et al., 2013). The amount of usable images for reconstruction of annual mass balance based on EOSS is more predictable with SAR imagery compared to using optical satellite data, due to a reliable repeat passes of the Sentinel-1

satellites and transparent clouds.

Wet snow absorbs most of the microwave signal, returning little energy back to the sensor (Stiles and Ulaby, 1980). The roughness of snow/ice and incidence angle of the SAR satellite sensor also affects the backscatter signal strength (Shi and Dozier, 1995). In both the Sentinel-1A and Radarsat-2 backscatter time-series an abrupt seasonal change was found, from warm and wet conditions in the end of the melt season to the onset of the cold season with dry and cold conditions

(corresponding to no.1 in Fig. 2). These backscatter differences are present due to a change in weather conditions between 26 August 2015 (DOY 238) and 7 September 2015 (DOY 250) causing an increase of ~ -20 dB to ~ -7 dB (Fig. 2). A temperature record from the closest weather station in Ny-Ålesund (8 m a.s.l.) shows lower temperature in the period between the images, indicating even colder temperatures on the glaciers since they are located on higher elevation (Fig. A2). In addition, an optical image from 9 September 2015 (DOY 252) over Kongsvegen shows significant precipitation as snow

(Table A5), that also indicates conditions < 0 degrees ºC on the glacier. The snowline, where the onset of cold season starts, corresponds to number 8 in Fig. 2, representing the EOSS. The onset of surface melt is characterized by a rapid decrease in backscatter values (corresponding to no.6 in Fig. 2, Stiles and Ulaby, 1980; Smith et al., 1997; Wolken et al., 2009), most likely introduced by warm and wet weather conditions (Rotschky et al., 2011). The lowest backscatter values in the ablation zone are found when wet snow covers the glacier (corresponding to no. 9, the blue color in Fig. 2), which also reflects the

length of the melt season. This has until now been mostly studied using QuikSCAT data with daily temporal resolution, but low spatial resolution (e.g. Rotschky et al., 2011). It is difficult to define the melt season accurately in time from Radarsat-2 images due to the low repeat time of 24-days (Fig. 5). Sentinel-1 provides 6-days repeat time with two satellite sensors, and can be used to study the length of the melt season more accurately in terms of spatio-temporal resolution.

The TSL migrates up-glacier dependent on elevation (corresponding to no.7 in Fig. 2). It is also possible to track the

wetness in the snow up-glacier with the surface dry-to-wet seasonal snow line (corresponding to no.10 in Fig.2), but here we have focused on the TSL. During the melt season, seasonal snow melts away exposing a rough ice or firn surface, thus creating a sharp contrast to the smooth and wet snow. Based on similar observations by Hall and others (2000), we believe this can be observed from SAR backscatter, due to different roughness lengths between the two targets causing higher scattering of the ice surface (Fig. 3). We have used this difference in roughness to track the TSL within the melt season.

When tracking the TSLs within a melt season, the few optical images available can be gap-filled with SAR-data (Fig. 4ab). TSLs from optical and SAR images acquired almost at the same day were manually selected from the glaciers (Table A5). Heatplots, a DEM, SAR backscatter and optical satellite images were used for visual inspection of the TSL positions in the analysis (see also temperature plot in Fig. A1). Using mean backscatter in elevation zones of 25 m, gives a location of the TSL representing the width of the glacier, compared to using a centerline representing one point on the glacier. The same





visual interpretation was used to retrieve the EOSS from SAR acquisitions on four of the glaciers in 2015 and 2016 (Hellstugubreen, Storbreen, Nigardsbreen and Austdalsbreen). On Gråsubreen we detected a melt regime which did not correlate with elevation, as the snow melt was first apparent in the convex areas higher up on the glacier, and not in the lower part where the snow accumulates (most likely due to the local topography and wind patterns).

5         Strong correlation was found between TSLs from Sentinel-2 and Landsat 8 satellite images and TSLs from Sentinel-1 images (Fig. 4abc). Snow events lower the backscatter values significantly during the melt season, due to a smoother surface and high absorption of radar waves as the snow is wet. An example of this was when lower backscatter values were observed in the ablation area of for example on Hellstugubreen in 3 October 2016 (DOY 277), most likely due to wet snow lowering the roughness and absorbing microwave energy (Fig. 4a). Snow was observed in an optical image on 6

October 2016 (DOY 280) on Hellstugubreen (not shown). In addition, yet a different example from Kongsvegen, a snow event caused an outlier on this glacier, which happened between the acquisition dates of the compared optical and SAR satellite images (7-9 September 2015, DOY 250 to 252, respectively, see outlier in fig. 4c). The TSL elevation was 475 m on the SAR image (acquired 2 days before the optical image) and 100 m on the optical image due to new snow on the glacier tongue.

15         The EOSS and ELA in Fig. 4d reflects the maritime-continental transect of glaciers in southern Norway, as the most continental glaciers have the highest ELA. EOSS positions from SAR backscatter data correlated well with ELA calculated from the SMB-gradients, in addition to in situ measurements (Fig. 4d). This indicates that EOSS derived from Sentinel-1A and B SAR data may be used as input when reconstructing annual mass balance series.

        In both study regions, we found a discrepancy between the datasets in that the TSLs from SAR generally show a 3

% lower altitude compared to optical derived TSLs. On Kongsvegen, as the snowline retreats up into the superimposed ice or firn area, the roughness difference is less as compared with glacier ice, making it more difficult to retrieve a correct TSL (Fig. 5 and 6a). Generally, TSLs are visually more apparent on optical images than on SAR backscatter images, but it may still be challenging to derive the TSL when superimposed ice is present (e.g. Winther, 1993; Kundu and Chakraborty, 2015).

        The glacier geometry, elevation, size, sun exposure, and local snow accumulation influences the melt pattern of

seasonal snow on glaciers, and thus the EOSS. If the glacier spans a small elevation range, the TSL could be more difficult to trace, the TSL may be either above or below the elevation range that the glacier cover. In addition, interaction between snow lines and crevassed areas, especially in ice falls, can be a challenge on larger glaciers (e.g. Chinn, 1995). Extracting TSLs and EOSS using SAR backscatter data can be valuable for three purposes. 1) Studying the melt regime of well-studied glaciers with already existing surface mass balance programs. 2) Including many glaciers in the analysis, if the purpose is to

derive the TSL and EOSS from SAR-imagery to reconstruct annual mass balance series regionally and retrieve a statistical robust result. 3) To use TSLs and EOSS for calibration and validation of mass balance models (e.g. Hulth et al., 2013).



## 5.2 Identifying glacier facies

SAR backscatter imagery can identify SAR glacier zones corresponding to glacier facies (Fahnestock et al., 1993; Brown et al., 1999; Rau et al., 2000; König et al., 2002; Engeset et al., 2002; Jaenicke et al., 2006; Langley et al., 2008), because the SAR backscatter is influenced by physical properties of ice and snow, despite weather conditions and surface texture (e.g. Smith et al., 1997; Rau et al., 2000). In this application scenario, we have observed glacier facies from 2009-2016 including time-series of both Radatsat-2 (24-day repeat) and Sentinel-1A (12-day repeat). Previous studies have found distinct glacier zones on Kongsvegen (Engeset et al., 2002; König et al., 2004; Brandt et al., 2008; Langley et al., 2008), and these zones also corresponds to previous literature (Benson, 1962; Rau et al., 2000; Cuffey and Paterson, 2010). The dry-snow SAR-zone is absent on Kongsvegen and Holtedahlfonna (Engeset et al., 2002; Langley et al., 2007). The firn line does not vary much from year to year, however several years of negative mass balance will eventually migrate the firn line up-glacier, and vice versa with positive mass balance years (e.g. König et al., 2004; Brown, 2012). Here, we have examined glacier facies using SAR backscatter time-series on Kongsvegen and Holtedahlfonna.

Wet snow typically has the lowest backscatter values, followed by wet and dry ice (here, these show similar values), dry superimposed ice and dry snow/firn (Fahnestock et al., 1993). A dry snow pack has low dielectric contrast, and SAR waves are volume scattered. In the firn area, ice lenses, pipes and layers are acting as randomly oriented dielectric cylinders, and are responsible for the high scattering of microwave signal back to the SAR sensor (Woodhouse, 2006). The backscatter response of superimposed ice is dependent on air bubble content and size, where high frequency of bubbles typically causes higher backscatter values. The superimposed ice zone has lower backscatter response compared to the firn in the cold season, but higher backscatter than glacier ice, as the ice transmits much of the microwaves (e.g. Langley et al., 2009) and additionally the ice surface are controlled by roughness (Shi and Dozier, 1995; Hall et al., 2000).

The SAR-zones vary seasonally as backscatter changes from being sensitive to surface properties in the melt season to volume properties in the cold season (Fig. 2). On Kongsvegen we found the SAR glacier zones (corresponding to no.5 in Fig. 2): 1) frozen-percolation zone (Firn zone), 2) wet-snow zone, 3) ice zone. In addition, we have included a fourth SAR-zone, 4) the superimposed ice (SI zone). The firn and SI zones are part of the accumulation area, and the ice zone is part of the ablation area. The wet-snow zone represents wet snow and firn in the melt season, but also rain events during the cold season (Sect. 5.3), and was found in both the ablation and accumulation area.

In Fig. 6a we show three distinct SAR-zones on Kongsvegen, approximately around the elevations 0-500 m, 500-700 m and 700-800 m in the cold season (7 September 2015 to 4 May 2016, DOY 250 to 125). These SAR-zones mirror the glacier facies, ice, SI and firn area, respectively. Stabilization of the SAR-zones between years in the cold season indicates established glacier facies in time (Fig. 5 and 6a). Thus, the Radarsat-2 time-series from 2009–2015 on Kongsvegen showed a relatively stable firn line altitude and a superimposed ice altitude (SIA), even though a retreat of SIA can be observed in 2011 and 2012 (Fig. 5). The winter SAR images are useful for identifying the superimposed ice zone since with optical satellite imagery it is hard to separate this zone from the bare ice zone (e.g. Kundu and Chakraborty, 2015). Surface snow,



lacking large scatterers and dielectric contrasts promotes low backscatter; glacier ice below the surface snow cover promotes surface scattering at the snow-ice interface and transmission with the ice volume. Together these scattering processes generally result in low backscatter intensity (light blue to yellow colors in Fig. 6a). The backscatter signal from the ice is changing seasonally to changes in surface roughness, as melt water is creating a rougher ice surface due to local melt water

streams (Shi and Dozier, 1995; Hall et al., 2000) (Fig. 6a, see yellow to orange colors on ice in the melt season, compared to the light blue color in the cold season).

Glacier zones are less clear on Holtedahlfonna, compared with Kongsvegen. Backscatter values were found to be gradually increasing between 570 m and 775 m, but no clear separation between zones was found on Holtedahlfonna (Fig. 7a). The lower part of Holtedahlfonna, Kronebreen, had several stripes (below 570 m in the heatplot in Fig. 7a) of high

backscatter, indicating highly crevassed areas due to double-bounce scattering of the physical corner reflector (Woodhouse, 2016). Although such corner reflectors probably blurred the seasonal signal, there was still seasonal variability with low backscatter values (blue colors) in early spring 2016 (around 28 May, DOY 149) indicating the onset of melt season (Fig. 7a).

Previously, one satellite image was used when mapping glacier zones from SAR imagery. Here, we examined

glacier zones on Kongsvegen and Holtedahlfonna thoroughly through time, using consecutive and dense time-series of SAR imagery, which therefore improves the potential to detect and study changes in facies. The results show a stabilization of the zones in the cold season, and one can therefore be more certain about the designation of glacier facies.

**5.3 Firn evolution and internal processes**

For the first time, we present a comparison of Sentinel-1A backscatter time-series with results from a surface mass balance

and firn pack evolution model. A coupled energy balance – multi-layer firn model was used to simulate the evolution of temperature, density and water content in snow and firn. The model results were based on Van Pelt and Kohler (2015), in which the mass balance and firn evolution of Kongsvegen and Holtedahlfonna are studied. Van Pelt and Kohler (2015) used downscaled HIRLAM regional climate model output. In this work, the climate forcing was updated, since the Ny-Ålesund weather station data were used as sea-level forcing for temperature, relative humidity, cloud cover, precipitation and air

pressure. Elevation lapse rates for temperature and precipitation were optimized to remove biases between modeled and observed seasonal mass balance. The precipitation lapse rate was optimized against winter balance data, and the temperature lapse rate was optimized against summer balance data. Lapse rates were determined individually for Kongsvegen and Holtedahlfonna, which is mainly relevant for precipitation, which is known to increase much faster with elevation on Kongsvegen than on Holtedahlfonna (Nuth et al., 2012). Model output contains daily, depth-dependent fields with a 300 m

horizontal spacing along centerline profiles on the glaciers. The sub-surface grid contains a total of 100 layers with layer thickness increasing with depth and ranging between 10 and 40 cm. Several modeled outputs are directly or indirectly related to wetness. Here, firn air-content and the subsurface water content were compared with the SAR backscatter time-series. Firn air-content (m) is a measure of the amount of empty pore-space in a vertical column (in this study we used a 2 m column).



Since ice has zero pore-space, the firn air-content is a direct measure of snow/firn depth and density (Fig. 6b). The subsurface water content (kg m$^{-3}$) represents irreducible water at a specified depth. From the subsurface water content variable in the snow/firn, we have calculated depth (in meter) of where the transition between wet (water content > 0) and dry snow/firn (water content = 0), indicating which depth has a water content above zero (Fig. 7b). This result was correlated

with the SAR backscatter data.

In the cold season, the ice surface covered with seasonal snow had low firn air-content in the beginning of this period, and increasing in time and towards the firn area, as the presence of snow accumulates up-glacier (e.g. at 23 April 2015, DOY 113 in Fig. 6b). We observed high backscatter values (Fig. 6a) when firn air-content was high (Fig. 6b), as the dry and cold conditions were found to enhance volume scattering. For each Sentinel-1A acquisition, we found a positive

significant correlation between firn air-content and backscatter in the cold season, and negative significant correlation in the melt season (Fig. 6c). The reason for the negative correlation is the change of surface properties affecting the backscatter signal, as the volume scattering in the cold season is exchanged with the SAR sensitivity to wet snow and higher surface roughness in the melt season. The point in Fig. 6c on 1 September 2016 (DOY 245) showed no correlation, because it is located in the transition zone between the melt season and cold season, where multiple melt end freeze events happen within

short time spans. The snow line was identified in the modeled data (Fig. 6b), as well as in in the observed backscatter data (Sect. 5.1).

In the beginning of the cold season when the surface refreezes, backscatter values are sensitive to subsurface melt in the snow and firn pack (Ashcraft and Long, 2006, Rotschky et al., 2011). A spatio-temporal refreeze signal from the Sentinel-1A time-series was observed in the firn area of, both on Kongsvegen and Holtedahlfonna (corresponding to no.2 in

Fig. 2). This was also observed as a weaker signal in the Radarsat-2 time-series just after the melt season each year (Fig. 5 and Fig. A3). Christianson and others (2015) found a perennial water aquifer in the upper part of the firn area on Holtedahlfonna, and they argued that the firn aquifer had a depth of approximately 3.5 to 15 meters. We found stabilizing backscatter intensity values in time during the winter period (Fig. 6a and 7a) and speculate that C-band SAR backscatter data does not receive backscatter responses from firn that deep as presented by Chrisianson and others (2015). An aquifer, or wet

conditions in the firn, causes absorption limiting backscatter response from the depth of the firn volume (e.g. Ashcraft and Long, 2006). In the firn zone of the glaciers (> 650 m on Kongsvegen and > 775 m on Holtedahlfonna), we found a gradual increasing backscatter signal during the cold season from 7 September to 12 December 2015 (DOY 250 to 346) (Fig. 6a and 7a, ~ -10 dB to ~ -4 dB). This is most likely due to multiple scattering events, from volume scattering in the snow and firn, in addition to wetness in the firn that gradually refreezes due to the winter cold wave. This can reflect stored and percolated

water in the firn from melt and rain events that refreeze over time, also releasing latent heat possibly slowing down the process. Backscatter eventually stabilized (12 December 2015, DOY 346, in Fig. 7a) and showed similar high values throughout the rest of the cold season. The backscatter measured at the sensor antenna are the sum of multiple scattering occurrences throughout the firn volume. The radar waves penetration depth varies over time and contributes to the backscatter, and the backscatter intensity is therefore not constant in time and represent the integral of depth in time.  Using



the modeled data, we were able to explore the penetration of SAR backscatter in snow and firn. Due to absorption of backscatter values in wet snow and firn conditions, we were able to estimate the depths of the intersection between the dry and wet zone in the firn pack (Fig. 7b). Results indicate increased penetration depths over time as a significant correlation between the modeled depth of the subsurface dry-to-wet transition zone and the backscatter values for the uppermost part of

the glacier (Fig. 7c). The deeper the dry-to-wet transition zone, the higher SAR backscatter values indicating a potential linear relationship (Fig. 7e).

The black points in Fig. 7a were selected along the low to high backscatter transition in the firn as an indication of the dry-to-wet zone (as explained above). The mean transition depth of the point values from Fig. 7b is 1.7 m when all 9 points were included. When only including the upper 6 points with significant correlation > 920 m (as showed in Fig.7c and

e), a mean transition depth of 2.0 m was found. We speculate that backscatter intensity can not identify small changes in snow when conditions are dry during wintertime due to high volume scattering.

Modeled data were here used to explain backscatter time-series data. This might be inverted in the future, as SAR backscatter data will be further understood. Such information is valuable in remote regions in the high Arctic that are lacking meteorological stations and where it is costly to do field observations.

**5.4 Weather events on glaciers in the cold season**

High Arctic regions like Svalbard, have encountered substantial warming the last decades, especially in the winter season with a temperature increase of 3.8 ºC from the measured period 2001–2015 to the reference period 1971–2000 (Isaksen et al., 2016). Frequent warm winter weather events are occurring especially related to increasing numbers of melt and precipitation days in the mid-winter (Vikhamar-Schuler et al., 2016). Winter rain events are important to map in a

glaciological context as it can be a substantial component to internal accumulation as it refreezes in the snow pack (e.g. Jansson et al., 2003; Van Pelt and Kohler, 2015, Van Pelt et al., 2016). The SAR backscatter contrast of ice and snow is directly dependent on the meteorological conditions before and at the time of acquisition of the satellite image. Rain on an ice surface gives stable backscatter values similar to dry ice, since glacier ice is still rough even when wet. Consequently, roughness is considered as a highly important scattering variable (e.g. Hall et al., 2000). Wet snow absorbs SAR-waves, and

little energy is transmitted back to the SAR sensor in comparison to a dry snow and firn surface (Rott, 1984). During dry snow conditions, the radar signals depend on the underlying firn or ice conditions (e.g. Brown et al., 2005). In this application scenario, we have examined a rain event on Kongsvegen and Holtedahlfonna from dense Sentinel-1A images during the cold season (corresponding to no. 3 in Fig. 2). Radarsat-2 ScanSAR satellite images were used to investigate the extent of the rain event.

On 5 January 2016 (DOY 5), Sentinel-1A acquired an image over the Kongsfjorden region indicating wet conditions on the upper part of Kongsvegen and its surroundings (corresponding to no. 3 in Fig. 2). The temperature and precipitation data from Ny-Ålesund showed a warm and wet weather event from 29 December 2015 (DOY 363) to 4 January 2016 (DOY 4) (Fig. A2 and Table A4). Fig. 8a and b shows backscatter images before and after the rain event, and Fig. 8c,





which shows the difference between the two, indicates an area of wet conditions (blue color). Presumably, the extent of the rain event covered the entire surface of Kongsvegen. However, on the lower part on the glacier (the ice facies) this was hard to observe due to little snow and rough surface. A clearer signal was found further up-glacier with present seasonal snow on firn (white circle in Fig. 8b).

Wet conditions in a SAR image (indicated by low backscatter values in Fig. 8b) do not necessary mirror the weather conditions at the time of acquisition of the satellite image, but may instead reflect the previous consecutive days, since rainwater percolates through the snow and firn pack creating prolonged wet conditions. In Radarsat-2 ScanSAR data (Fig. 9) and meteorological data (Fig. A2 and Table A4), colder conditions were measured on January 4 DOY 2016 (DOY 4, one day before the Sentinel-1A acquisition). As stable low backscatter values were observed on 4-5 January 2016 (DOY 4 to 5) (Fig.

9 and Table A4), we suggest that capillary water in the snow and firn, which was not yet frozen, were detected by the Sentinel-1 image on 5 January 2016 (DOY 5). This could be related to the isolation capability of snow and release of latent heat when water refreezes.

The same wet snow conditions were not found on Holtedahlfonna (Fig.7a and 8c). We observed a rain event on Holtedahlfonna on 30 December 2015 (white arrow on DOY 364 in Fig. 9), corresponding to 26.5 mm rain in Ny-Ålesund

(Table A4). In the following days, Holtedahlfonna had gradually higher SAR backscatter values, while Kongsvegen remained dark (low backscatter). Precipitation either fell as snow on Holtedahlfonna between 3-5 January 2016 (DOY 3 to 5), or conditions were cold with no snowfall. Thus, the Radarsat-2 ScanSAR images revealed different local weather conditions in the Kongsfjorden region. Colder conditions are in general present on Holtedahlfonna as it has a more continental climate compared to Kongsvegen, indicating that the rain water in the snow and firn pack froze faster than on

Kongsvegen.

Directly after the winter rain event, we observed a changing backscatter pattern in the ablation zone and lower part of the superimposed ice zone indicating change in snow and ice properties on Kongsvegen (corresponding to no. 4 in Fig. 2). A change was also apparent from the modeled firn air-content in Fig. 6b, and reflects an increase in pore space in the snow and firn pack after a rain event. Backscatter values increased with > 2 dB after the winter rain event between 300 – 600 m,

and the border between the superimposed ice and ablation zone became unclear (Fig. 6a). We know from *in situ* observations on Kongsvegen that glacier ice in the ablation area was covered with little snow in the spring 2016. The lower zone of the ablation area might have had little snow cover already before the rain event since the backscatter was continuously low during winter (indicated by light blue color in Fig. 6a. Present around the 200 m elevation line from 17 January to 4 May 2016 (DOY 17 to DOY 125). In the upper zone of the ablation area, deeper snow depth was present before the rain event. Ice

lenses and pipes might thus have been created from this penetration of water in the snow after the rain event, resulting in higher permittivity of the SAR signals showed by increasing backscatter values (indicated by yellow color in Fig. 6a. Present around the 400 m elevation line from 17 January to 4 May 2016, DOY 17 to DOY 125).

Detection of winter rain events from SAR time-series might be used to refine modeling inputs, especially in regions where meteorological data are scarce.



## 5.5 Assistance by SAR data in glacier outline mapping

It is well known that glacier ice is mapped more efficiently with optical satellite imagery than with SAR satellite data (e.g. Shi et al., 1994). However, when using the multispectral band ratio method on a single optical image for deriving glacier outlines (e.g. Paul and Kääb, 2005), it can be difficult to separate seasonal snow from glacier and perennial snow patches

(e.g. Andreassen et al., 2012; Winsvold et al., 2016). Local differences in weather conditions within a single optical satellite scene are common in high mountain regions. Mapping conditions in a single optical image might not be ideal due to cloud cover, seasonal snow around the glacier perimeter, and thin layers of newly fallen snow. Interferometric coherence images have been used for deriving glacier outlines (e.g. Atwood et al., 2010; Falk et al., 2016). In maritime regions such as south-western Norway it can be challenging to use SAR coherence images, due to rapid coherence loss between the time of

acquisitions often caused by precipitation, melt and wind. SAR backscatter information can be a valuable replacement. Wet snow and ice surfaces have lower backscatter values than a snow-free surface outside the glacier (Rott, 1984; Strozzi et al., 1997). In this application scenario, we investigate how averaged summer SAR images from glaciers in southern Norway can be used to assist in glacier outline mapping (e.g. corresponding to no. 11 in Fig. 2).

In 2015, seasonal snow remained throughout the ablation season in our study area (Fig. 10d, e and f). A SAR

composite consisting of the mean of 6 summer SAR-images might assist the mapping process of glacier outlines and perennial snow patches (Fig. 10a, b and c). This is possible as the seasonal snow was less visible in the summer SAR images, compared with the optical image (Fig. 10b and e, and c and f). Most likely, some SAR wave penetration in the seasonal snow is possible despite the potentially wet conditions. Thin snow that contaminates the optical mapping scene might become largely transparent in the SAR stack. In addition, we speculate that water content in the seasonal snow might have been low

at the days and time-of-day of SAR acquisitions. The Sentinel-1 scenes used here are taken in the afternoon at time 15:45. Finally, water might drain easier from snow patches than on the glacier surface due to often higher slopes, which results in a clearer signal from on-glacier pixels since the signal is consistently low.

For our example, it is possible to retrieve a good estimate of the glacier outlines when summer SAR images are averaged. Although the backscatter images were affected by some radar distortions, we found good agreement with the

existing glacier outlines (Andreassen et al., 2008) and the Landsat 8 image (Fig. 10a and d). The potential of the stacked SAR image can be further improved with double revisit time of Sentinel-1A and B. Clearly, the SAR image stack used cannot fully replace optical scenes for glacier mapping, but be of help as an additional layer to discriminate glacier areas from seasonal snow.

## 6 Concluding remarks

In this paper, we have analyzed temporal trends and stack statistics from SAR backscatter time-series over glaciers using Sentinel-1A and B and Radarsat-2 images. Sentinel-1 provides free and open data of higher nominal revisit time than any SAR instrument earlier. The time-series is consistent, making it possible to retrieve detailed information about the glacier





surface and subsurface in time. We have focused on the variable pattern of backscatter values on glaciers, and not the absolute values. Further work is needed for developing standardized semi-automatic or automatic methods, where the backscatter coefficient gamma nought should be used as input, for a complete geometric and radiometric processing of the SAR images. Still, standardized threshold values on the backscatter coefficients might be difficult to apply due to difference

in glacier dynamics and climate, and difference in polarization, incidence angle, ascending or descending paths, and the applied processing algorithm. On a regional basis, well-known classification regimes can be used to outline the glacier mapping variables from SAR data (Kääb et al., 2014).

Using five application scenarios, we presented new insights on how to exploit dense SAR data for glacier mapping purposes. We validated and compared our results with model data, meteorological data, existing glacier outlines from optical
data, surface mass balance data, and remote sensing data (Radarsat-2 ScanSAR, Sentinel-2 and Landsat 8).

- We have demonstrated the possibility of tracking TSLs during the melt season and deriving the EOSS from Sentinel-1A and B backscatter time-series. TSL data were found to be valuable for regionally extrapolating and estimating annual mass balance in areas without *in situ* measurements. Even though the temporal resolution of optical imagery will
increase with the sister satellite Sentinel-2B in orbit, maritime regions will remain cloudy and hinder dense time-series of high- to medium resolution optical imagery, and SAR time-series can therefore act as a data gap-filler. Additionally, Sentinel-1 time-series can be used to measure snow melt parameters on glaciers with high spatial resolution and with 6-days temporal resolution (i.e. dry-to-wet snow line, onset of melt season and length of the melt season).

- Glacier SAR-zones corresponding to glacier facies were observed from the backscatter time-series. Time-series from
2009-2016 using Radarsat-2 and Sentinel-1A SAR backscatter data, showed relatively stable glacier facies on Kongsvegen and Holtedahlfonna. Dense SAR time-series have a potential for more accurate delineation of SAR-zones, compared to using only one acquisition as in previous studies.

- We presented a descriptive comparison of modeled surface and firn evolution patterns with SAR backscatter time-series. The penetration depths of Sentinel-1A backscatter values in the firn are not constant in time, and resembled modeled
results. Strong correlation exist between the modeled firn air-content and SAR backscatter values throughout the whole year. Strong correlation was also found between the modeled depth of the subsurface dry-to-wet conditions in the firn pack and winter SAR backscatter values. Our findings are important to understand glaciological processes, and we have shown the potential to combine modeled snow/firn evolution and SAR backscatter data match each other.

- Winter weather events are predicted to be more frequent in the Arctic in the future. If rain water from winter weather
events refreezes in the firn area, it will contribute to internal accumulation of the glacier. In dense Sentinel-1A backscatter time-series, it was possible to detect such winter rain events in the accumulation areas of glaciers, even when the rain event happened before the satellite acquisition.



- It can be challenging to map glacier outlines from the multi-spectral band ratio method when a thin layer of new snow or much seasonal snow is present in the optical mapping scene. With averaged summer SAR backscatter images, we showed a potential for assisting the glacier mapping process.

5      With 6-day repeat cycles (Sentinel-1A and B) even more variability of glacier conditions will be captured, e.g. detecting winter rain events more thoroughly, or tracking the end of summer snowline more precisely. Even though optical imagery often is preferred for many glacier mapping approaches because it measures in similar wavelengths as our eyes, SAR backscatter has the potential for being increasingly applied to map glaciers. SAR backscatter time-series can be used as a refined modelling input, especially in regions where surface mass balance and meteorological data are scarce. However, 10   more investigations are needed for deriving robust end products.

**Appendices**

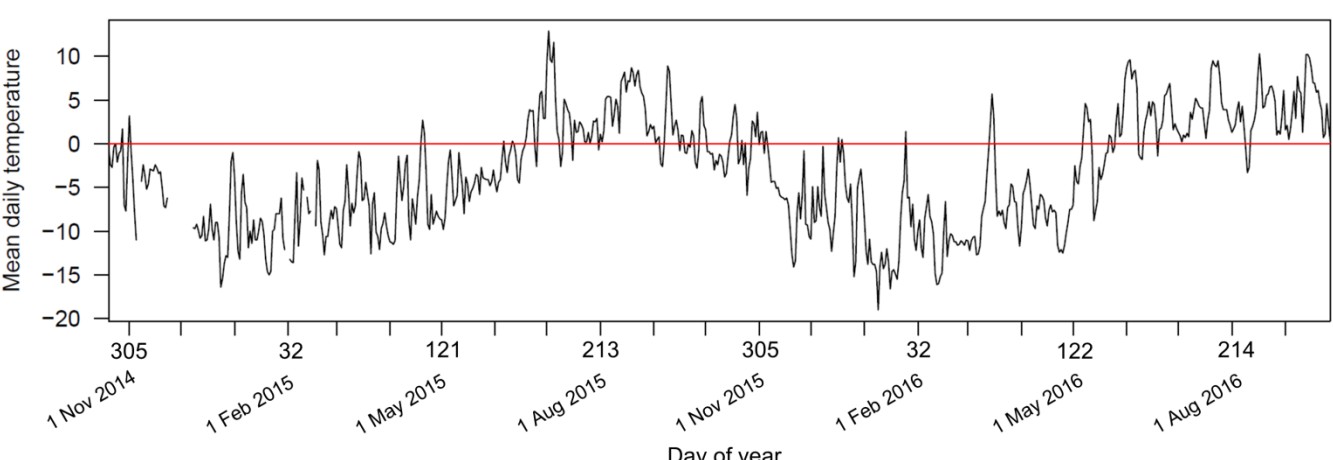

**Fig. A1:** Temperature record from Juvvasshøe meteorological station (ID 15270, 1894 m a.s.l.) from October 2014 to October 2016 (downloaded from eKlima.no, 2016).



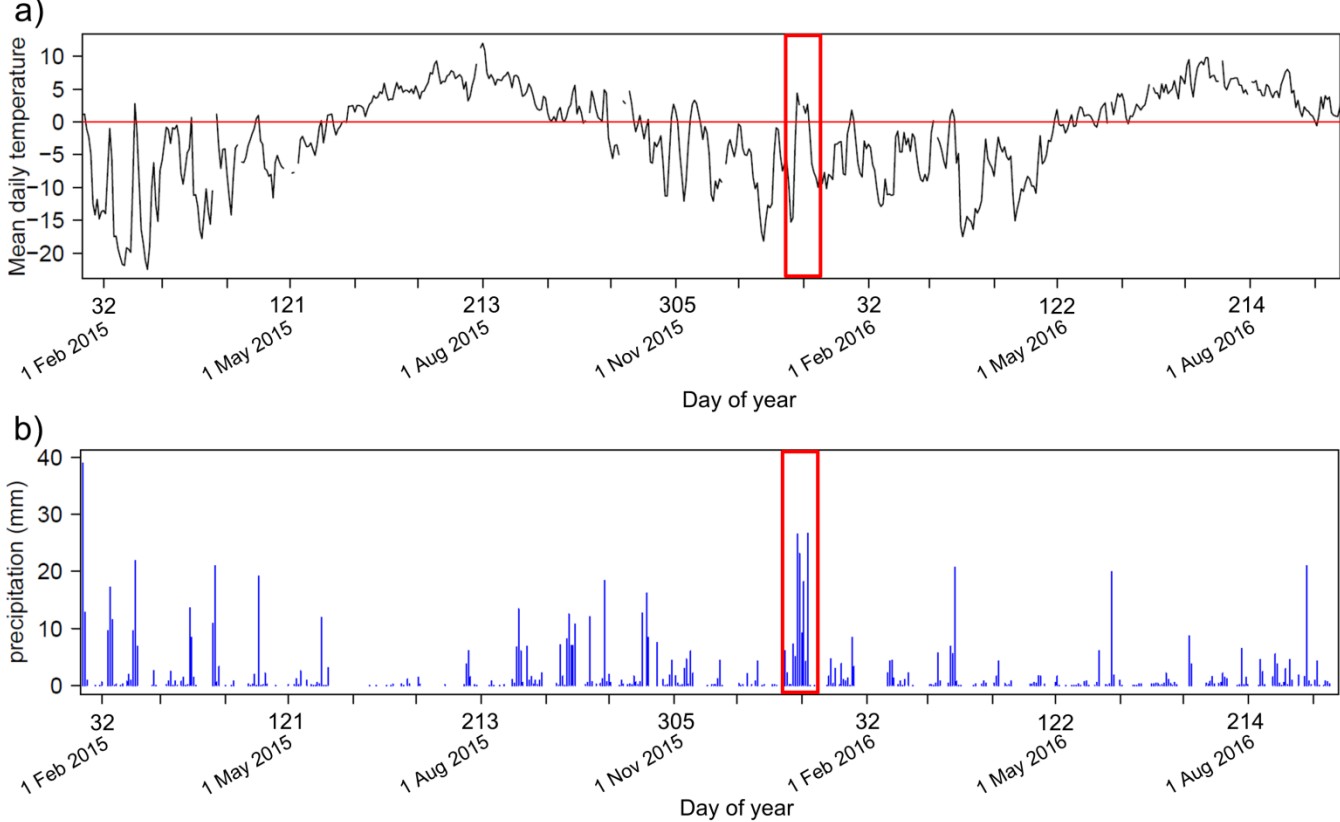

**Fig. A2: (a)** Temperature, and **(b)** precipitation record from the meteorological station at Ny-Ålesund (ID 99910, 8 m a.s.l.) from 22 January 2015 to 13 September 2016 (DOY 22 2015 to 257 2016). The red box indicates the time of the winter rain event with wet and warm conditions, triggering low backscatter intensity of the snow and firn in upper parts of Kongsvegen on the 5 January 2016 SAR image. After the rain event, it might been snow fall on the glacier (downloaded from eKlima.no, 2016).



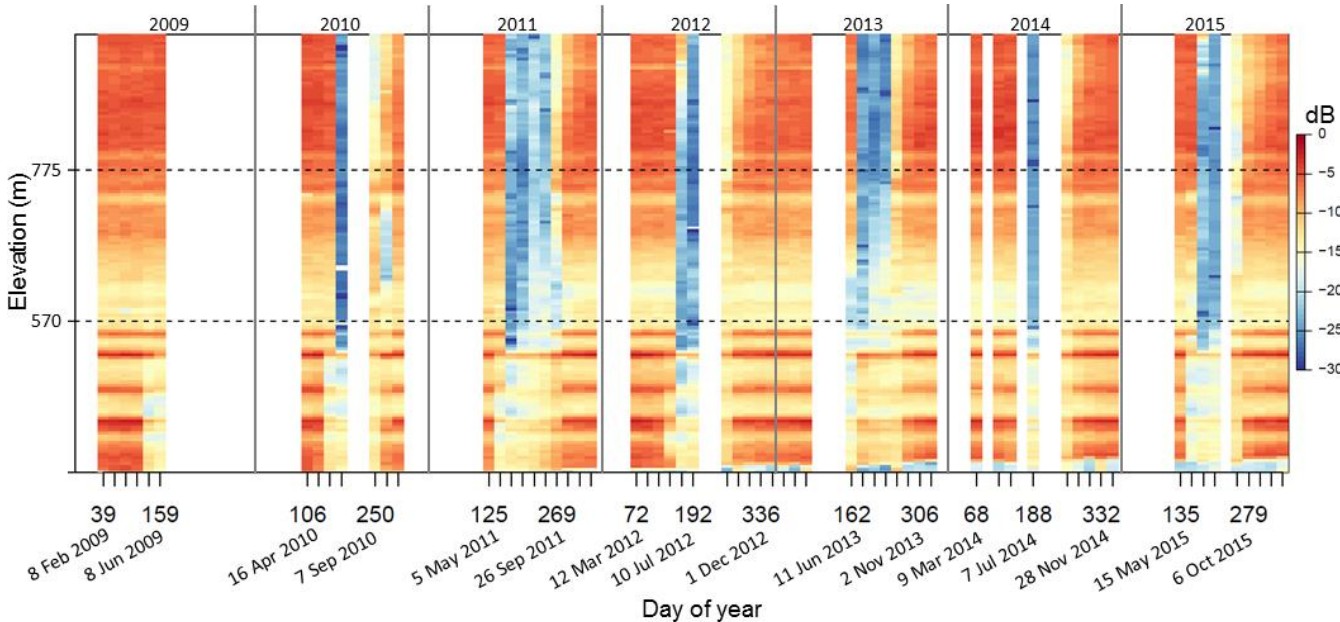

**Fig. A3:** Radarsat-2 time-series of SAR backscatter values (dB) along a centerline profile on Holtedahlfonna from 2009 to 2015. The Radarsat-2 time-series has a lower temporal resolution than the Sentinel-1A time-series (24-day vs. 12-day repeat cycles, respectively). Despite the time gaps in the Radarsat-2 time-series, the refreezing signal in the upper part of the firn area/zone, is similar to the Sentinel-1 time-series (Fig.7a). This is shown by a gradual increase in backscatter in the firn zone right after the melt season, when the winter cold wave penetrates the firn and stabilizes it. Time is plotted as Julian days on the x-axis (day of year).





**Table A4:** Measured temperature (TAM=mean, TAN=minimum, TAX=maximum), precipitation (mm) (RR) and snow depth (cm) (SA) from the Ny-Ålesund meteorological station, in the period between two acquisitions of Sentinel-1A data (24 December 2015 to 5 January 2016). Precipitation and maximum temperature marked in blue and red might have contributed to wet and warm conditions on the glaciers. (x = No Data).

| Date | DOY | TAM | TAN | TAX | RR | SA |
|---|---|---|---|---|---|---|
| 24 Dec 2015 | 258 | -5.8 | -6.9 | -4.7 | 6 | 20 |
| 25 Dec 2015 | 259 | -9.5 | -11.2 | -5.3 | 2.2 | 20 |
| 26 Dec 2015 | 260 | -15.3 | -16.4 | -11 | 0.1 | 9 |
| 27 Dec 2015 | 261 | -14.6 | -17 | -14.2 | 0.1 | 13 |
| 28 Dec 2015 | 262 | -3.7 | -14.5 | -2.8 | 7.2 | 18 |
| 29 Dec 2015 | 263 | 4.4 | -3.2 | 6.2 | 5 | 12 |
| 30 Dec 2015 | 264 | 2.6 | -0.1 | 7.2 | 26.5 | 5 |
| 31 Dec 2015 | 265 | x | 1.2 | 7.4 | 23.1 | x |
| 1 Jan 2016 | 1 | 2.4 | -0.1 | 5.2 | 9.2 | x |
| 2 Jan 2016 | 2 | 1.3 | -0.4 | 3.5 | 18.2 | 4 |
| 3 Jan 2016 | 3 | 2.7 | 0.6 | 5.6 | 4.2 | x |
| 4 Jan 2016 | 4 | -1.4 | -2.9 | 1.2 | 26.7 | x |
| 5 Jan 2016 | 5 | -6.4 | -8.1 | -2.7 | 0 | x |

**Table A5:** Transient snow lines at Hellstugubreen and Kongsvegen derived from Sentinel-2A. Sentinel-2A (S2A in orange), Landsat 8 (L8 in green) and Sentinel-1A (S1A in black). Gray shade indicates images from 2016.

|  | Optical (S2A and L8) | | | SAR (S1A) | | |
|---|---|---|---|---|---|---|
|  | DATE | DOY | TSL (m) | DATE | DOY | TSL (m) |
| **Hellstugubreen** | 17 Aug 2015 | 229 | 1600 | 16 Aug 2015 | 228 | 1575 |
|  | 18 Aug 2015 | 230 | 1600 | 16 Aug 2015 | 228 | 1575 |
|  | 11 Sep 2015 | 254 | 1750 | 9 Sep 2015 | 252 | 1800 |
|  | 19 Aug 2016 | 232 | 1775 | 22 Aug 2016 | 235 | 1800 |
|  | 4 Sep 2016 | 248 | 1800 | 3 Sep 2016 | 247 | 1825 |
|  | 20 Sep 2016 | 264 | 1875 | 15 Sep 2016 | 259 | 1875 |
| **Kongsvegen** | 1 Aug 2015 | 213 | 500 | 2 Aug 2015 | 214 | 475 |
|  | 13 Aug 2015 | 225 | 575 | 14 Aug 2015 | 226 | 525 |
|  | 22 Aug 2015 | 234 | 600 | 26 Aug 2015 | 238 | 575 |
|  | 9 Sep 2015 | 252 | 100 | 7 Sep 2015 | 250 | 475 |
|  | 18 Sep 2015 | 261 | 625 | 19 Sep 2015 | 262 | NA |
|  | 2 Jul 2016 | 184 | 400 | 3 Jul 2016 | 185 | 375 |
|  | 9 Jul 2016 | 191 | 475 | 15 Jul 2016 | 197 | 475 |
|  | 2 Aug 2016 | 215 | 675 | 27 Jul 2016 | 209 | 575 |
|  | 10 Aug 2016 | 223 | 700 | 8 Aug 2016 | 221 | 650 |

*Author contributions*

S.H.Winsvold developed the concepts of the study together with A.Kääb. and C.Nuth. The Sentinel-1 and 2 and Landsat data were geoprocessed and analysed by S.H.Winsvold. W.V.Pelt modelled the firn air content and water content on Holtedalsfonna and Kongsvegen in Svalbard. T.Schellenberger processed the Radarsat-2 data, and C. Nuth processed
Sentinel-1 for comparison tests, both using GAMMA. L.M. Andreassen provided surface mass balance data from NVE and helped with result interpretations. S.H.Winsvold prepared all figures and tables, and wrote the manuscript. All authors contributed on editing the paper.

*Competing interests*

The authors declare that they have no conflict of interest

*Acknowledgement*

The study was in parts funded by the European Research Council under the European Union's Seventh Framework Programme (FP/2007–2013)/ERC grant agreement No. 320816, the ESA project Glaciers_cci (4000109873/14/I-NB) and by the Norwegian Space Centre contract NIT.06.15.5. We are very grateful to ESA for provision of the Copernicus Sentinel-1 and Sentinel-2 data. Radarsat-2 Wide Fine Mode data were provided by NSC/KSAT under the Norwegian-Canadian
Radarsat agreements 2007–2015. Landsat imagery were provided by the U.S. Geological Survey through Earth Explorer. The Tandem Intermediate DEM was produced and provided by the German Space Agency (DLR) under proposal # IDEM_GLAC0425. The Norwegian mapping agency for provision of their DEM. ASTER GDEM [v2] is a product of NASA and METI. Thanks to I. Brown and S. Wunderle for their comments on the manuscript when being opponents on Winsvold's PhD-defence. Furthermore, thanks to K. Langley, T. Dunse and B. Altena for helpful discussions, and finally
thanks to P.M. Lefeuvre and R. McNabb help with R-coding and batch scripting, respectively.

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



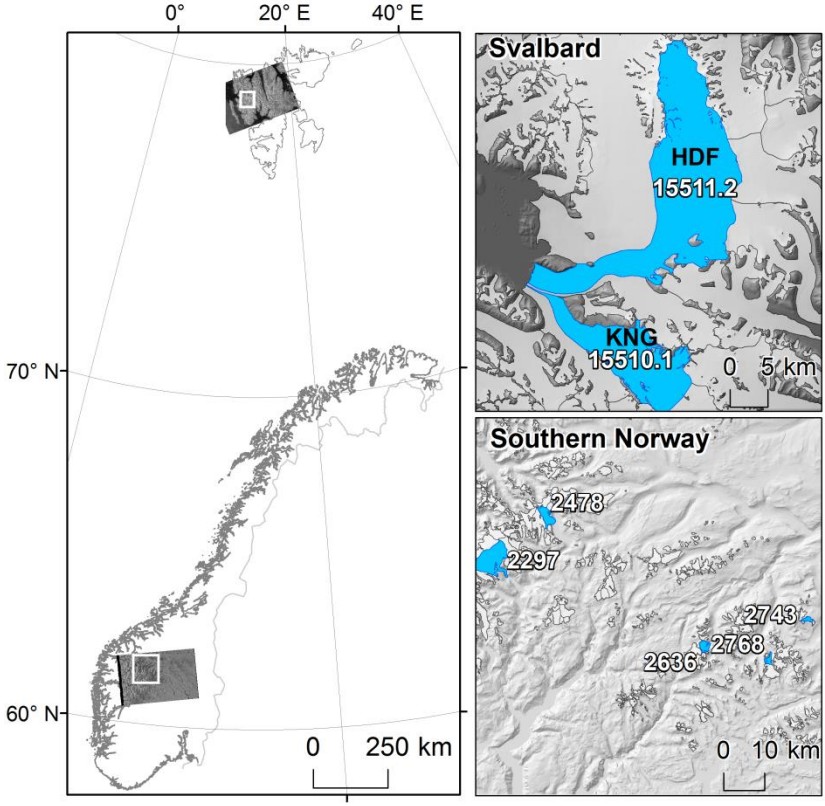

**Figure 1:** Study area in Svalbard (Local IDs: 15511.2=Holtedahlfonna and 15510.1=Kongsvegen) and southern Norway (Local IDs: 2478=Austdalsbreen, 2297=Nigardsbreen, 2636=Storbreen, 2768=Hellstugubreen, and 2743=Gråsubreen) (Glacier outlines from Nuth et al., 2013, Andreassen et al., 2008 and Paul et al,. 2011).





**Figure 2:** A heatplot of Sentinel-1A backscatter (dB) time-series from 22 January 2015 to 13 September 2016 (DOY 22 2015 to 257 2016, along a profile on Kongsvegen, Svalbard. Time is shown as Julian days on the x-axis (day of year), with 12-days between each acquisition. The numbers represent 11 possible glacier mapping variables that can be detected from the SAR time-series, 1) onset of cold season, 2) freeze-up and evolution of the firn area as the winter cold wave penetrates the snow and firn, 3) winter rain event, 4) change of surface properties after winter rain event, 5) glacier facies (based on definitions from Rau et al., 2000; Cuffey and Paterson, 2010), separated by firn line altitude (FLA) and superimposed ice altitude (SIA) (SI=superimposed ice), 6) onset of melt season, 7) transient snow lines (TSL) illustrated by the difference between ice and wet snow, 8) end of summer snowline (EOSS), an estimation of the equilibrium line altitude (ELA), 9) length of melt season, 10) surface dry-to-wet snow line, 11) glacier outline or calving front. The two sketches above the heatplot represent the SAR-zones in the cold season (winter with dry and cold conditions; snowpack in white due to high volume scattering) and in the melt season (summer with warm and wet conditions), together representing a full mass balance year. (Illustration insets above the heatplot are modified based on de Ruyter de Wildt, 2002).





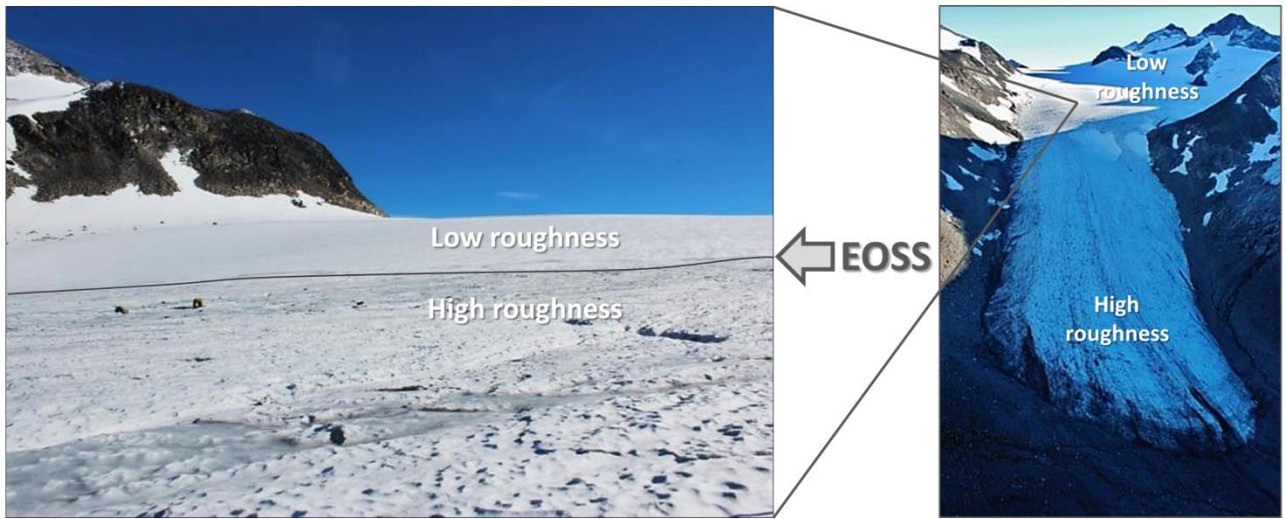

**Figure 3:** Photos illustrating roughness differences between ice/firn and seasonal snow. The images of Hellstugubreen were acquired on 12 September 2016 (DOY 256), when the end of summer snow line can be observed, often corresponding to the ELA (corresponding to no.8 in Fig. 2). Photos: Liss M. Andreassen, NVE.







**Figure 4: (a)** Time-series of Sentinel-1A and B backscatter images where mean dB-values have been calculated for elevation zones of 25 m (y-axis). Numbers on x-axis are the dates (month and day, MMDD). White lines indicate missing Sentinel-1A scenes. Grey lines separate between years. The stripe of low backscatter data in the upper part is due to areas in the radar shadow. Note: 3-15 October 2016 (DOY 277 to 289 are Sentinel-1B acquisitions with 6-days repeat time **(b)** Elevation of TSL from optical satellite imagery, corresponding to the numbers in a). Sentinel-2A on 18 August 2015 (DOY 230), the rest is Landsat 8 images. Green points and boxes are *in situ* EOSS derived from hand held GPS. **(c)** Manually picked TSLs from optical and SAR images acquired almost at the same day (Table A5). The maximum difference in acquisition dates between sensors is ± 6 days. **(d)** Plot of EOSS from SAR and ELA from the surface mass balance (SMB) gradients and field observations (Pearson's correlation coefficient: 0.92, p-value: 0.00015). Austdalsbreen (A) is an outlier most likely due to new snow causing uncertainties in the measured SMB, since no stakes were found at higher elevation. The maximum discrepancy between a field observation and a SAR acquisition were 21 days (SAR acquisition 3 October 2016 (DOY 277) and in situ 12 September 2016 (DOY 256) on Hellstugubreen). Abbrevations: S = Storbreen, H = Hellstugubreen, G = Gråsubreen and N = Nigardsbreen.





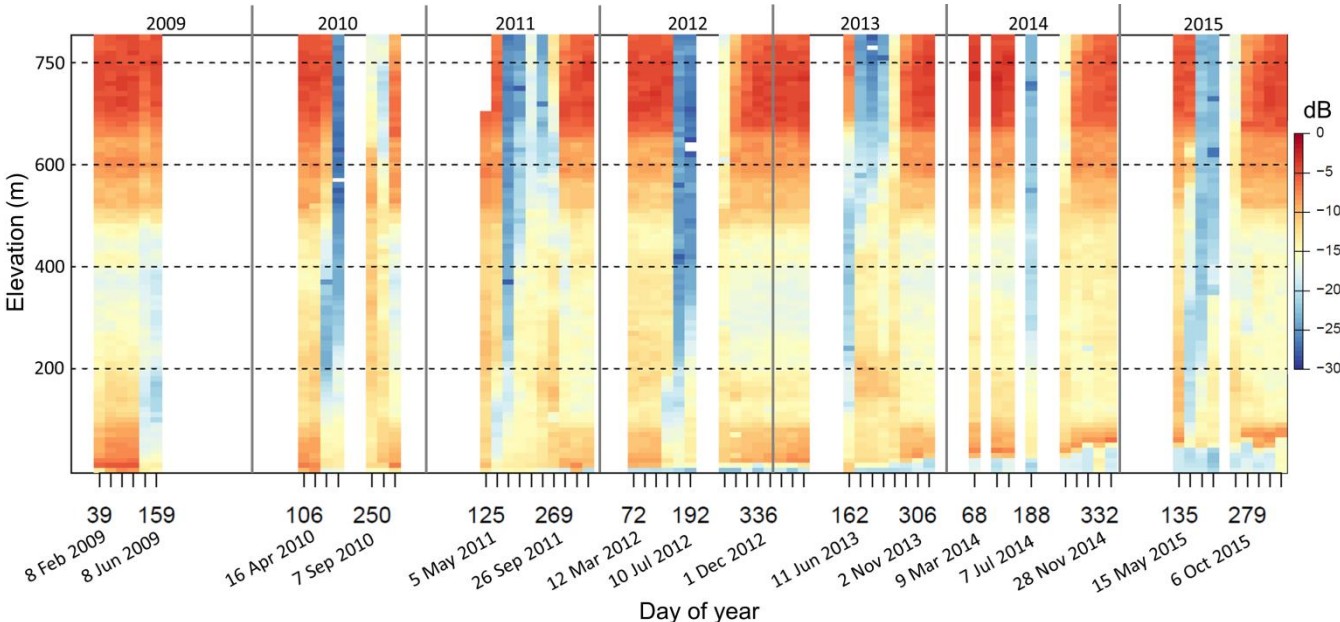

**Figure 5:** Radarsat-2 time-series of SAR backscatter values (dB) along a centerline profile on Kongsvegen from 2009 to 2015. An abrupt change in the start and end of melting season (e.g. in year 2012 and 2013) was observed. The Radarsat-2 time-series has a lower temporal resolution than the Sentinel-1A time-series (24-day vs. 12-day repeat cycles, respectively, see Fig. 6a). The backscattering values are in decibel (dB). Time is shown as Julian days on the x-axis (day of year), and dates are plotted for assistance.









**Figure 6:** Observed backscatter and modeled values along the Kongsvegen centerline profile. **(a)** Sentinel-1A backscatter (dB) time-series from 22 January 2015 to 13 September 2016 (DOY 22 2015 to 257 2016). **(b)** Modeled firn air-content in a vertical column of 2 m (The colorbar is in m w.e.), **(c)** Spearman correlation between the firn air-content (in b) and backscatter values (in a) along the centerline for each Sentinel-1A acquisition time (each 300 m point). Significant correlation values (after Bonferroni multiple testing correction) are shown in red. Points on the black stippled line corresponds to missing data (white areas in figure a). For a) and b) time is shown as Julian days on the x-axis (day of year), and dates are plotted for assistance.









**Figure 7** The figures show results from Holtedahlfonna **(a)** Sentinel-1A SAR backscatter (dB) time-series from 22 January 2015 to 13 September 2016 (DOY 22 2015 to 257 2016). Black points indicate where the backscatter stabilizes in the cold season. **(b)** Daily modeled water content for each 300 m point along a centerline profile and the transition depth (in m) between wet snow/firn (Water content > 0) and dry snow/firn (Water content = 0). The white regions in the plot indicates no water since it is glacier ice with no snow in the summer, or with dry snow conditions in the winter. Transition depth of 0 m indicates wet snow conditions on the surface. **(c)** Spearman correlation between depth of dry-to-wet transition zone and backscatter values plotted with elevation (y-axis) in firn area above 775 m. We used a time period with stable conditions: 7 September 2015 to 22 April 2016 (DOY 250 2015 to 113 2016. Black box in a and b). Significant correlation values (after Bonferroni multiple testing correction) are plotted in red. **(d)** Plot of backscatter values (dB) vs. dry-to-wet transition depth (layer) for points in c) with p-value > 0.05 (Points under ~920 m, using the same time period 7 September 2015 to 22 April 2016, DOY 250 2015 to 113 2016). The correlation coefficients are 0.266 (Pearson's correlation coefficient) and 0.3 (Spearman rho). **(e)** Plot of backscatter values (dB) vs. dry-to-wet transition depth (layer) for points in c) with p-value < 0.05 (Points above ~920 m, using the same time period as in c) and d) ). The correlation coefficients are 0.834 (Pearson's correlation coefficient) and 0.725 (Spearman rho). For a) and b) time is shown as Julian days on the x-axis (day of year), and dates are plotted for assistance.



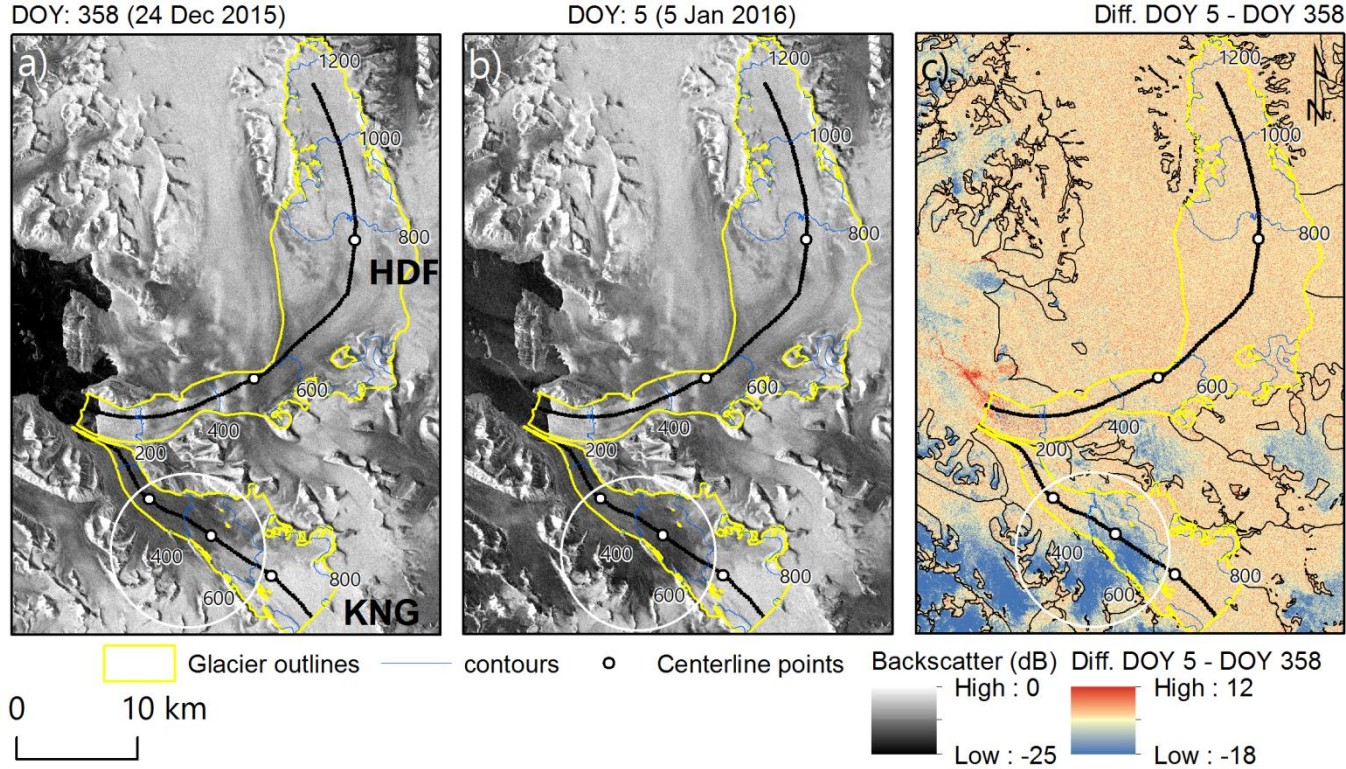

**Figure 8: (a)** Sentinel-1A backscatter (dB) image from 24 December 2015 (DOY 358), and **(b)** the following Sentinel-1A backscatter image from 5 January 2016 (DOY 5) showing remnants of the rain event. **(c)** The difference between 5 January 2016 (DOY 5) and 24 December 2015 (DOY 358) SAR intensity images. Blue color indicates a lowering of backscatter values between the images, showing wetter conditions in the upper firn of Kongsvegen (white circles). Glacier outlines from 2007 (Nuth et al., 2013).



**Figure 9:** Radarsat-2 ScanSAR mode backscatter (dB) images from 29-30 December 2015 (DOY 363 and 364) and 3-5 January 2016 (DOY 3 to 5). We use Radarsat-2 ScanSAR to explain the extent of the winter weather event. Conditions on Holtedahlfonna were wet on 30 December (see white arrows), but were dry and cold again on 3-5 January 2016 (DOY 3 to 5) (higher backscatter than on 30 December 2015 (DOY 364). Low backscatter values were found on Kongsvegen from 3-5 January 2016 (DOY 3 to 5. See red arrows), indicating wetter conditions and more rainfall in this period compared to Holtedahlfonna.



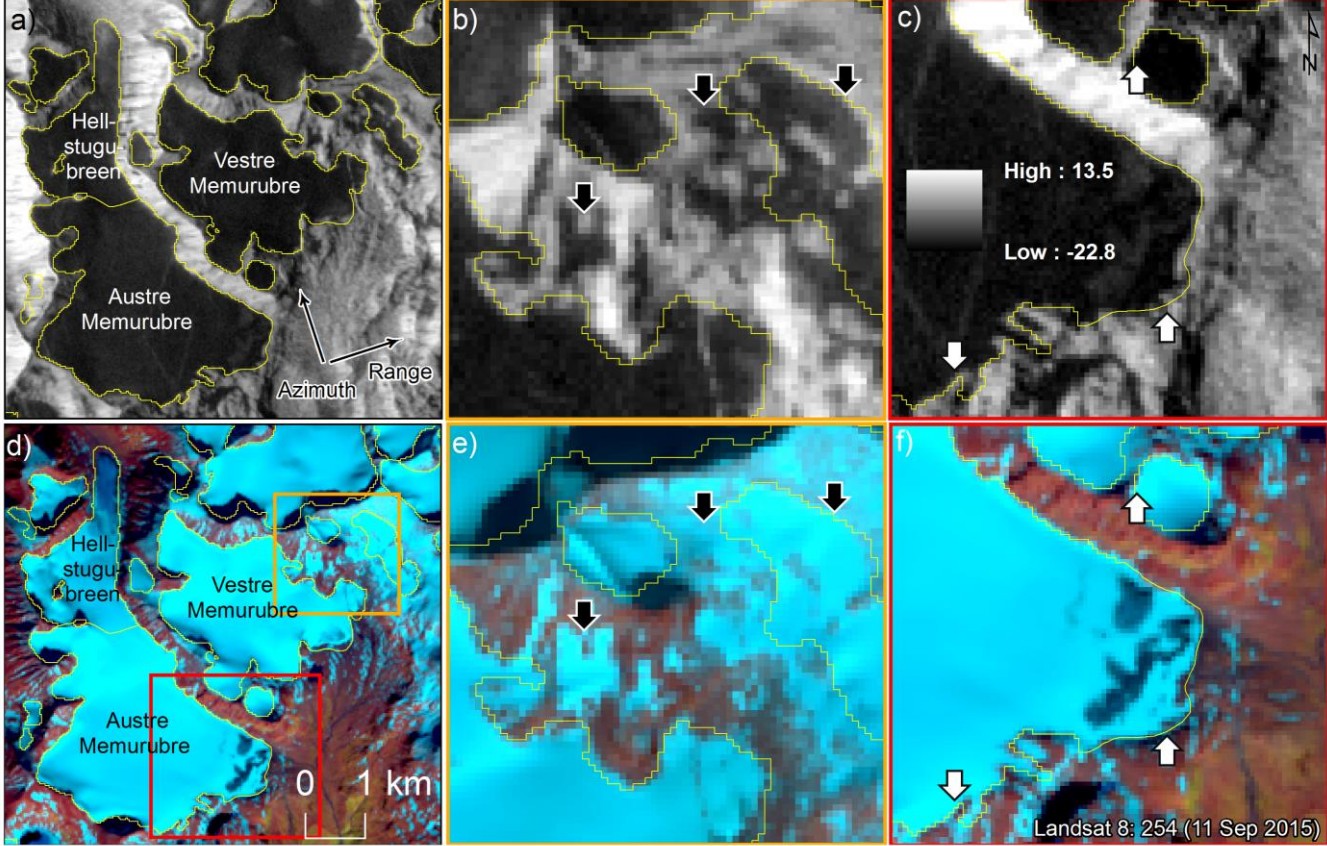

**Figure 10:** Glacier outline mapping for glaciers around Hellstugubreen. The inset shows zoom in on Austre Memurubre **(a)** Mean of 6 SAR backscatter (dB) images in the melt season from 23 July to 21 September 2015 (DOY 204 to 264). **(b)(c)** subsets of a) as indicated by the yellow and red rectangle in d). Azimuth angle = -18,3 degrees. **(d)(e)(f)** Landsat 8 OLI image 11 September 2015 (DOY 254) and subsets with poor glacier mapping conditions, since seasonal snow persisted around the glacier perimeters, and a thin layer of new snow as in b) and e). Black arrows in b) and e) and white c) and f), indicate places where a backscatter composite image can assist optical image when mapping glacier and perennial snow patches. Glacier outlines from 2003 (Andreassen et al., 2008), and therefore it is some discrepancy between the imagery presented here.