# Peer review of "Using SAR satellite data time-series for regional glacier mapping"

_The Cryosphere, 2017_

## Referee Comment (RC1) · Anonymous Referee #1 · 4 Oct 2017

Review of "Using SAR satellite data time-series for regional glacier mapping" By S. Winsvold and others

Summary

The authors present and analyze trends and statistics from Sentinel-1A and B and Radarsat-2 Synthetic Aperture Radar (SAR) backscatter data time-series over glaciers in arctic and sub-arctic Norway. The premise of the paper is that dense, high-resolution SAR satellite data time-series can lead to improved mapping of time-variable surface and subsurface glacier properties and features. The authors present results from five separate, but related analyses of SAR backscatter times-series that focus on variable patterns of backscatter change, including transient snow line mapping, glacier facies mapping, firn evolution, detection and extent mapping of winter rain-on-snow events,

and mapping glacier extent.

The paper acknowledges and utilizes the extensive background of research in using microwave remote sensing to map time-varying glacier properties, and extends this topic by utilizing higher-resolution SAR satellite data with faster revisit times. The authors illustrate an enriched characterization of glacier properties at the catchment scale resulting from the improved spatial and temporal resolution of SAR satellite data, and present an interesting comparison of SAR backscatter data time-series with modeled firn air and moisture levels, which is not entirely convincing without ground validation. Overall, the manuscript is well organized, but it requires some attention is specific areas to improve writing, phrasing, and organization in order to facilitate reader understanding and comprehension. I have some general comments, as well as a number of targeted comments and questions detailed below.

General Comments

The authors make very good use of the data. Plots are informative with necessary explanatory support. In the absence of in situ snow/firn data, all possible influences on radar energy must be addressed or explained with reference to relevant previous studies; authors should augment interpretations accordingly.

Targeted Comments

P1 L11: Suggest rephrasing "…used descriptive methods for outlining the possibilities…"

P1 L20: Eliminate "Finally"

P1 L23-25: This final statement of the abstract should be re-written to be more specific to this study. In its current form it can apply to nearly all glacier remote sensing studies. Also, "semi-automatically" is probably more appropriate at this stage.

P2 L27: Suggest adding the descriptive modifier "high-resolution" to "…dense SAR satellite image time-series…" as this reinforces the case that this is "new potential"

P4 L4: Given the specific applications of the data from these sensors, it would be helpful to remind the reader of the wavelength.

P5 L23: Consider describing the process of "multi-looked" rather than using it as a verb in this sentence.

P6 L5: Change to "...outdated, coarser...).

P7 L20: Readers would likely benefit from a reminder that QuikSCAT is a Ku-band sensor.

P7 L24: Re-phrase the first sentence of this paragraph as the statement in its current from is untrue or partially correct.

P8 L3: This is an interesting result, which adds credence to the use of dense, high-resolution SAR satellite data for this type of analysis. Other sensors used for this purpose would not have provided this level of detail in the melt patterns that are controlled by terrain parameters. It would be worth highlighting this.

P8 L6-7: Sentence should be re-written for clarity. Also, consider depth of snow as this can scale inversely with backscatter. Do you see this influence here?

P8 L10-14: This section of text needs to be re-written to achieve clarity.

P8 L17: Were the SMB gradients not determined through in situ measurements?

P8 L20: Hyphenate "...optical-derived"

P8 L22-23: This sentence needs clarification as the current writing is ambiguous.

P8 L28: List should be written in a consistent manner according to "valuable for" in the leading sentence. Consider colon usage and proper listing format.

P8 L28: More appropriately written as "Refining spatial variations in the melt pattern of well-studied..."

P9 L2 : Should be "SAR backscatter imagery can be used to..."

P9 L2: Second use of SAR is redundant.

P9 L2: There is inconsistent use of SAR glacier zones and glacier facies throughout the paper. Define these and strive for consistency in usage as it will be difficult for the readers if this isn't addressed. It will be important to start with a clarification about what is being classified and what is interpreted from these classifications. For example, "SAR backscatter imagery can be used to identify distinct zones of consistent backscatter levels that correspond to glacier facies. . ." Is there ambiguity in the classifications? Do SAR glacier zones directly relate to glacier facies? If so, the two terms should be treated separately, if not, they should be referred to as glacier facies.

P9 L7-8: Consider re-phrasing to clarify. "Zones" don't correspond to previous literature, rather to previous interpretations found in the literature.

P9 L17: How do you know this?

P9 L8: Just a clarification: The dry-snow glacier facies is absent here and the interpretation of the SAR backscatter data are consistent with this repeated observation.

P9 L18-20: Should be re-written to make the message clearer. Also, be careful not to present conflicting information about the winter glacier-ice zone.

P9 L22 : Should be ". . . we found the following. . ."

P9 L 23: Delete "frozen."

P9 L 24: Why isn't superimposed ice part of the list in the previous sentence? How do you know it isn't a saturation facies within the firn zone?

P9 L 27: Why not keep the reader directed at Fig. 2 instead of Fig. 6a?

P9 L 28-29: This might be best explained/defined at the beginning of this section as mentioned above.

P9 L29: Perhaps this is a better lead sentence for this paragraph.

[Figure]

P9 L33: Is this image combination available to add as a figure? If so, this would provide useful evidence for the chosen facies interpretation.

P10 L3-6: Consider re-phrasing this sentence. "…changing seasonally in response to changes…" Need to qualify or provide some scale for "local melt water streams" relevant to the frequency of the sensor used. Also consider the drying effect of melt water channeling and how that impacts the backscatter.

P10 L23: Updated from what to what?

P10 L25: Delete "elevation"

P10 L26-27: Please reword for clarification. Optimized – not "against." I'm assuming you mean "for"?

P10 L32: Please clarify "Several modeled outputs are directly or indirectly related to wetness." How does this relate to the previous sentence?

P11 L1: How is firn air content a direct measure of anything if it is a model simulation product?

P11 L3: delete "where"

P11 L6: This introductory sentence needs to be rewritten for clarity.

P11 L17-34: Good discussion. It would be worth including in the discussion, the size definition of a target that is considered transparent, compared to those considered good specular or diffuse reflectors at the c-band frequency.

P11 L21: Re-word: A perennial firn aquifer was found containing liquid water.

P11 L22: Clarification required. Is the firn aquifer depth 3.5-15 m, or is it located below the surface 3.5-15 m?

P12 L1: Backscatter values can't be absorbed, please reword/rephrase.

P12 L1-6: This contradicts the explanatory statements starting at P11 L28. How can

you be certain that SAR penetration depth is increasing? This needs to be reconciled, along with an explanation of possible contributing parallel processes occurring in an evolving snowpack exposed to a penetrating winter cold wave.

P12 L7-14: Suggest reorganization of this section to make it easier for the reader to follow. Also, include a transitional/introductory sentence to link the previous paragraph to this.

P12 L10: Modify. "... intensity can not 'be used to' identify..."

P12 L25: Change last word to 'surfaces'

P13 L23: Explain why an increase in pore space is occurred.

P13 L24: Delete 'with'

P13 L28: Case change 'p'

P13 L29: Rephrase sentence, change 'deeper snow depth'

P13 L29-32: This section needs a better explanation of the processes involved to support the conclusions. Why 'higher permittivity'? Could there be another explanation for a sharp increase in backscatter? Snow saturation and superimposed ice is common with these events – are these identifiable?

P15 L17: Rearrange wording – 'glaciers with high spatial resolution'

P15 L21: 'SAR-zones' should be 'glacier facies'

P15 L27-28: Rearrange/rephrase for reader clarity.

P17 L5: Rephrase

P29: Minor - Consider switching the color blocks for 'Cold season' and 'Warm season' to reflect customary warm and cool color designations.

---

## Referee Comment (RC2) · Anonymous Referee #2 · 25 Oct 2017

1. Does the paper address relevant scientific questions within the scope of TC? -> Yes; it does without question.

2. Does the paper present novel concepts, ideas, tools, or data? -> The innovation is the application and interpretation of time series of Sentinal SAR data in conjunction with Landsat 8 optical and RADARSAT SAR to five scenarios for Glacial morphological analysis. As such, it presents a benchmark for future analysis of any one or a combination of time series scenarios. The important contribution is the layout of the procedure for as well as the assessment of different approaches to mapping glacier morphology and process from satellite time series.

3. Are substantial conclusions reached? -> Whereas comparable assessments of new imaging technology might conclude with a 'capability' type of assessment, in this

study there is clearly a considerable depth of knowledge in the derivation, assessment and interpretation of the results that lends considerable confidence to the rigor of the conclusions. One would like to see more discussion of the results, given the insight learned, but this would result is a considerably longer paper. I hope the authors will pursue such work for the individual scenarios.

4. Are the scientific methods and assumptions valid and clearly outlined? -> Yes, well written and supported by prior literature. My only question is the use of parametric statistics, particularly the Spearman r correlation, for what might be a non-normal and probably not-independent sample (eg. Figure 6c). I believe the authors need to justify this.

5. Are the results sufficient to support the interpretations and conclusions? -> Yes. I do not see any interpretations that are not supported with the reported analysis and appropriate caveats.

6. Is the description of experiments and calculations sufficiently complete and precise to allow their reproduction by fellow scientists (traceability of results)? -> There is a great deal of analysis bound in this work and I have not traced all of the background references and their references. I believe that the literature cited will allow appropriate traceability.

7. Do the authors give proper credit to related work and clearly indicate their own new/original contribution? -> yes

8. Does the title clearly reflect the contents of the paper? -> The title is short and could benefit from the mention of the five scenarios which is a major contribution.

9. Does the abstract provide a concise and complete summary? -> Yes.

10. Is the overall presentation well structured and clear? -> Yes

11. Is the language fluent and precise? -> Some suggested corrections are provided below.

12. Are mathematical formulae, symbols, abbreviations, and units correctly defined and used? -> Yes

13. Should any parts of the paper (text, formulae, figures, tables) be clarified, reduced, combined, or eliminated? -> No

14. Are the number and quality of references appropriate? -> Yes

15. Is the amount and quality of supplementary material appropriate? -> Yes

Page 2, line 21, "...1980's have been..."

Page 7, line 24, "The TSL migration up-glacier is dependent..."

Page 8, line 8, "...in the ablation area of, for example, on..."

Page 8, line 10, "On addition, in a different example from..." (suggestion as 'yet' does not convey anything here)

Page 8, line 19: I suggest that "3% lower altitude" does not mean much as a percentage will change with absolute altitude. 3% of x scale or an absolute change of height would provide more clarity.

Page 10: Section 5.3 is a particularly strong contribution, in my opinion.

Page 14, line 20 "...at the days and time-of-day of SAR..." is awkward syntax. There may be a better construction.

Page 14, line27, "... but be of help..." perhaps should be "...but can be of help..."?

Page 15, line 29, I suggest that the authors mean "Winter rain events" as opposed to "Winter weather events" of which there are many in the Arctic and includes a wide range of precipitation types.

---

## Author Comment (AC1) · 6 Dec 2017

See supplement for both response letter and a marked-up manuscript version (PDF With track changes).

Please also note the supplement to this comment:
https://www.the-cryosphere-discuss.net/tc-2017-136/tc-2017-136-AC1-supplement.pdf

---

## Author Comment (AC2) · 6 Dec 2017

**The Cryosphere Discuss., https://doi.org/10.5194/tc-2017-136**
**Manuscript under review for journal The Cryosphere**
**Discussion started: 7 August 2017**

**Using SAR satellite data time-series for regional glacier mapping**

Solveig H. Winsvold, Andreas Kääb, Christopher Nuth, Liss M. Andreassen, Ward J.J. van Pelt, and
Thomas Schellenberger
* * *
**Final Response to Referee Comments**
**(Interactive Discussion)**
* * *
This is our response to the two referee comments published at TCD. In this letter, we reply to the referee comments on the points where a reply is requested (Authors answers in bullets). For the smaller corrections, such as spelling corrections, we state that we have done these by writing "**DONE**". A marked-up manuscript version (track changes in Word) is attached as a supplement in the response to referee comments.

Most important, we want to thank the two anonymous referees for their valuable, constructive and detailed comments. The comments will certainly improve the manuscript and are greatly acknowledged.

**Comments by Anonymous Referee #1:**

"The authors illustrate an enriched characterization of glacier properties at the catchment scale
resulting from the improved spatial and temporal resolution of SAR satellite data, and
present an interesting comparison of SAR backscatter data time-series with modeled
firn air and moisture levels, which is not entirely convincing without ground validation."

- We include a sentence and references about this under section 5.3: *"Validation of modeled subsurface density against shallow firn core observations on Kongsvegen and Holtedahlfonna has been discussed in Van Pelt and Kohler (2015), showing good agreement. There is no real validation for subsurface temperature and water content for these glaciers. But the model performed well in simulating vertical temperatures at the top of Lomonosovfonna, as shown in Van Pelt et al. (2014)."*

"Overall, the manuscript is well organized, but it requires some attention is specific
areas to improve writing, phrasing, and organization in order to facilitate reader understanding
and comprehension.»

- The text has been corrected and rewritten accordingly. In the revised manuscript changes are marked with track changes.

General Comments
The authors make very good use of the data. Plots are informative with necessary
explanatory support. In the absence of in situ snow/firn data, all possible influences
on radar energy must be addressed or explained with reference to relevant previous
studies; authors should augment interpretations accordingly.

- Agreed. We have had this in mind when correcting the manuscript, and some more references were added in line with the reviewers' feedback. See answers below.

Targeted Comments

P1 L11: Suggest rephrasing ". . .used descriptive methods for outlining the possibilities. . ."

- Agreed. The sentence has been rephrased to: *"On Sentinel-1A and Radarsat-2 backscatter time-series images over mainland Norway and Svalbard, we outline how to map glaciers using descriptive methods."*

P1 L20: Eliminate "Finally" **– DONE.**

P1 L23-25: This final statement of the abstract should be re-written to be more specific to this study. In its current form it can apply to nearly all glacier remote sensing studies. Also, "semi-automatically" is probably more appropriate at this stage.

- Agreed. We decided to remove the last sentence in the abstract as this is mentioned in the introduction and conclusion.

P2 L27: Suggest adding the descriptive modifier "high-resolution" to ". . .dense SAR satellite image time-series. . ." as this reinforces the case that this is "new potential" **– DONE.**

P4 L4: Given the specific applications of the data from these sensors, it would be helpful to remind the reader of the wavelength. "

- Agreed. It has been changed to: *"(center frequency of 5.405 GHz and wavelength of 5.5 cm)"*.

P5 L23: Consider describing the process of "multi-looked" rather than using it as a verb in this sentence.

- Agreed. We changed this sentence to*: "However, a multi-look algorithm was applied to reduce noise using spatial averaging."*

P6 L5: Change to ". . .outdated, coarser. . .)*. **– DONE.**

P7 L20: Readers would likely benefit from a reminder that QuikSCAT is a Ku-band sensor.

- Agreed. Changed to: *"This has until now been mostly studied using QuikSCAT data (a Ku-band sensor)…"*

P7 L24: Re-phrase the first sentence of this paragraph as the statement in its current from is untrue or partially correct.

- Agreed. We have clarified this by changing the sentence to: "*The TSL migration up-glacier is often correlated with temperature rise and topography"* (*e.g. Hall et al., 2000, corresponding to no.7 in Fig. 2)."*.

P8 L3: This is an interesting result, which adds credence to the use of dense, high resolution SAR satellite data for this type of analysis. Other sensors used for this purpose would not have provided this level of detail in the melt patterns that are controlled

by terrain parameters. It would be worth highlighting this.

- Thank you for pointing this out. We have included a sentence: *"On such glaciers, satellite sensors other than Sentinel-1A and B and their according dense time-series are not able to reveal the same level of temporal detail in melt patterns."*

P8 L6-7: Sentence should be re-written for clarity. Also, consider depth of snow as this can scale inversely with backscatter. Do you see this influence here?

- We believe that the wet snow absorbs much of the backscatter in this example, and little volume scattering can occur. The sentence is rewritten: *"In the melt season with warm and wet conditions, the backscatter signal after snow events might be reduced on the glacier. Firstly, they can cause a smoother surface compared to the underlying rougher ice surface, and, secondly, high absorption and forward scatter of the radar waves as the snow is wet (Rott, 1984)."*.

P8 L10-14: This section of text needs to be re-written to achieve clarity.

- Agreed. Rewrote the section: *"In addition, another example from Kongsvegen showed a snow event causing disagreement between the Landsat 8 image and Sentinel-1 images (see outlier in Fig. 4c and Table A5). This snow event must have happened between the acquisition dates of the optical and SAR satellite images compared (i.e., between 7-9 September 2015, DOY 250 to 252, respectively)."*.

P8 L17: Were the SMB gradients not determined through in situ measurements?

- Yes, that is correct. But the ELA is typically determined from the mass balance curves and not directly observed (e.g. measurements can be carried out after the first winter snow). There were also some in situ measurements of the TSL from field work. We changed the sentence to: *"EOSS positions from SAR backscatter data correlated well with ELA calculated from the SMB-curves that were derived from the in-situ measurements, in addition to direct in situ measurements of the EOSS."*

P8 L20: Hyphenate ". . .optical-derived" **– DONE.**

P8 L22-23: This sentence needs clarification as the current writing is ambiguous.

- Agreed. We have changed to: *"The same applies to optical satellite images as it is challenging to derive the TSL when superimposed ice is present (e.g. Winther, 1993; Kundu and Chakraborty, 2015)."*

P8 L28: List should be written in a consistent manner according to "valuable for" in the leading sentence. Consider colon usage and proper listing format.

- Agreed. These points are now listed.

P8 L28: More appropriately written as "Refining spatial variations in the melt pattern of well-studied. . ." **– DONE.**

P9 L2 : Should be "SAR backscatter imagery can be used to. . ." **– DONE.**

P9 L2: Second use of SAR is redundant.

- Agreed. Removed.

P9 L2: There is inconsistent use of SAR glacier zones and glacier facies throughout the paper. Define these and strive for consistency in usage as it will be difficult for the readers if this isn't addressed. It will be important to start with a clarification about what is being classified and what is interpreted from these classifications. For example, "SAR backscatter imagery can be used to identify distinct zones of consistent backscatter levels that correspond to glacier facies. . ." Is there ambiguity in the classifications? Do SAR glacier zones directly relate to glacier facies? If so, the two terms should be treated separately, if not, they should be referred to as glacier facies.

- We see that this is challenging, and we have rewritten the four first paragraphs in section 5.2 to clarify the used terms:

*"SAR backscatter imagery can be used to identify distinct zones of consistent backscatter that corresponds to glacier facies (Fahnestock et al., 1993; Brown et al., 1999; Rau et al., 2000; König et al., 2002; Engeset et al., 2002; Jaenicke et al., 2006; Langley et al., 2008).This is because the SAR backscatter is influenced by physical properties of ice and snow, albeit weather conditions and surface texture (e.g. Smith et al., 1997; Rau et al., 2000). Nonetheless, it is challenging to directly connect glacier facies to SAR classified glacier zones, as backscatter is a complex composite signal reflected from a surface volume, the material properties of which often vary temporally with external atmospheric forcings e.g. with winter rain events (Sect. 5.5). The term glacier facies are defined as properties on a glacier dividing one part of the glacier from others, often connected with mass balance processes (i.e. ablation and accumulation area), and glacier facies are equalized with the term glacier zones (Cogley et. al., 2011). Here we use the term glacier facies for snow and ice ground properties, and the term SAR glacier zones for the interpretations and classifications from SAR satellite imagery. SAR glacier zones often have an annual frequency, but also vary seasonally as backscatter changes from being sensitive to surface properties in the melt season to volume properties in the cold season (Fig. 2). We define the following SAR glacier zones (corresponding to no.5 in Fig. 2): 1) percolation zone (firn zone), 2) wet-snow zone, 3) ice zone, 4) the superimposed ice (SI zone) (e.g. Rau et al., 2000; Cuffey and Paterson, 2010). The firn and SI zones are part of the accumulation area, and the ice zone is part of the ablation area. The wet-snow zone represents wet snow and firn in the melt season (Sect. 5.1), but also rain events during the cold season (Sect. 5.5), and occurs in both ablation and accumulation areas.*

*In this application scenario, we present a time-series of SAR glacier zones from 2009-2016 including Radarsat-2 (24-day repeat) and Sentinel-1A (12-day repeat) images in northwest Svalbard. Previous studies have correlated distinct glacier facies with SAR glacier zones on Kongsvegen (Engeset et al., 2002; König et al., 2004; Brandt et al., 2008; Langley et al., 2008), and these facies also correspond to previous interpretations in literature (Benson, 1962; Rau et al., 2000; Cuffey and Paterson, 2010). The dry-snow glacier facies is absent on both Kongsvegen and Holtedahlfonna as melting occurs over the entire surface, (Engeset et al., 2002; Langley et al., 2007) which agrees with our interpretation of the SAR backscatter time-series (Fig. 2). The firn line does not vary much from year to year, however several years of negative mass balance will eventually migrate the firn line up-glacier, and vice versa with positive mass balance years (e.g. König et al., 2004; Brown, 2012).*

*Wet snow typically has the lowest backscatter values, followed by wet and dry ice (here, these show similar values, Fig. 2), dry superimposed ice and dry snow/firn (Fahnestock et al., 1993). A dry snow pack has low dielectric contrast, and SAR microwaves are volume scattered. In the firn area, ice lenses, pipes and layers act as randomly oriented dielectric cylinders, responsible for the high scattering of microwave signal back to the SAR sensor. We consider snow and firn to be Rayleigh scatterers when the particle size is lower*

*than 10% of the wavelength (λ = 5.5 cm) or 0.55 cm, and Mie scatterers up to 10 times the wavelength, or 55 cm (Woodhouse, 2006). The backscatter response of superimposed ice is dependent on air bubble content and size, where high frequency of bubbles typically causes higher backscatter values (e.g. König et al., 2002, Langley et al., 2009). Thus, during the cold season, the superimposed ice zone has typically lower backscatter compared to firn, but higher than glacier ice (e.g. Langley et al., 2009). During the melt season, high backscatter on ice surfaces is caused by surface roughness rather than volume scattering (Shi and Dozier, 1995; Hall et al., 2000)."*

P9 L7-8: Consider re-phrasing to clarify. "Zones" don't correspond to previous literature, rather to previous interpretations found in the literature.

- Agreed. We have changed the sentence to: *"…and these facies also correspond to previous interpretations in the literature".*

P9 L17: How do you know this?

- We are sorry that references were missing. We now refer to *"(e.g. König et al., 2002, Langley et al., 2009)"*.

P9 L8: Just a clarification: The dry-snow glacier facies is absent here and the interpretation of the SAR backscatter data are consistent with this repeated observation.

Agreed. We have clarified the sentence: *"The dry-snow glacier facies is absent on both Kongsvegen and Holtedahlfonna as melting occurs over the entire surface, (Engeset et al., 2002; Langley et al., 2007) which agrees with our interpretation of the SAR backscatter time-series (Fig. 2)."*

P9 L18-20: Should be re-written to make the message clearer. Also, be careful not to present conflicting information about the winter glacier-ice zone.

Agreed. See previous answer under P9 L2 (last paragraph).

-

P9 L22 : Should be ". . . we found the following. . ." **– DONE.**
P9 L 23: Delete "frozen." **– DONE.**

P9 L 24: Why isn't superimposed ice part of the list in the previous sentence? How do you know it isn't a saturation facies within the firn zone?

- Agreed. The SI-zone is now included in the list. We refer to SI based on previous literature (Engeset et al., 2002; König et al., 2004; Brandt et al., 2008; Langley et al., 2008).

P9 L 27: Why not keep the reader directed at Fig. 2 instead of Fig. 6a?

- Agreed. This has been changed.

P9 L 28-29: This might be best explained/defined at the beginning of this section as mentioned above.

- Agreed. We have rewritten the first part of section 5.2 to be more clear about SAR glacier zones and glacier facies.

P9 L29: Perhaps this is a better lead sentence for this paragraph.

- Agreed. This sentence is now first in this paragraph.

P9 L33: Is this image combination available to add as a figure? If so, this would provide useful evidence for the chosen facies interpretation.

- We've decided not to include a figure because it will not add much to the conclusion of section 5.2 about glacier facies. A figure showing the difference between SAR glacier zones on an optical and SAR image on Kongsvegen is found below. One needs to have a trained eye to see where the superimposed ice zone is in the optical image. With winter SAR image time-series it is possible to observe how the facies stabilizes in time (e.g. Fig. 2 and 5 in manuscript). This is more challenging to follow with optical imagery due to clouds and the spectral response of only the surface. We have chosen to modify the sentence about this to: *"The winter SAR images are useful for identifying the superimposed ice zone since with optical satellite imagery it can be challenging to separate this zone from the bare ice zone (e.g. Kundu and Chakraborty, 2015)."*

[Figure]

Figure 1: The figure shows comparison of glacier facies on Kongsvegen – Ice, superimposed ice and firn in the upper part. 1) An optical Landsat 8 image (enhanced histogram) and 2) a winter Sentinel-1 SAR image, in addition to 3) a mean backscatter image.

P10 L3-6: Consider re-phrasing this sentence. ". . .changing seasonally in response to changes. . ." Need to qualify or provide some scale for "local melt water streams" relevant to the frequency of the sensor used. Also consider the drying effect of melt water channeling and how that impacts the backscatter.

Agreed. We have clarified this: *"We suggest that backscatter amplitude from ice surfaces changes seasonally due to surface roughness variations, as melting creates a rougher ice surface due to changes in local topography on the ice surface e.g. micro water evacuation streams (Shi and Dozier, 1995; Hall et al., 2000) (Fig. 6a, see yellow to orange colors on ice in the melt season, compared to the light blue color in the cold season). According to the modified Rayleigh criterion a surface is considered rough in the C-*

*band SAR (using an incidence angle of 38 degrees) when the root-mean-square (rms) surface height variation is more than 1.3 cm, and considered a smooth surface when the rms height variation is less than 0.2 cm (e.g. Lillesand et al., 2004). Drying of melt water channels do not cause the backscatter to be higher in the cold season compared to the melt season because they cover only a very small percentage of the SAR pixels, therefore we believe the ablation effect (caused by melting and water on the ice surface) is responsible for a rougher surface and higher backscatter."*

P10 L23: Updated from what to what?

Agreed. This has been rewritten: *"The model has previously been used to simulate the long-term (1961-2012) mass balance and firn evolution of Kongsvegen and Holtedahlfonna, as described in Van Pelt and Kohler (2015). For the experiment in this paper the model is forced with weather station data from the Ny-Ålesund weather station (provided by the Norwegian Meteorological Institute), which provided time-series at sea-level for temperature, precipitation, relative humidity, cloud cover and air pressure."*

P10 L25: Delete "elevation" **– DONE.**
P10 L26-27: Please reword for clarification. Optimized – not "against." I'm assuming you mean "for"? **– DONE.**

P10 L32: Please clarify "Several modeled outputs are directly or indirectly related to wetness." How does this relate to the previous sentence?

- We have removed this sentence since it is not important for describing the data we use in this work.

P11 L1: How is firn air content a direct measure of anything if it is a model simulation product?

- We have removed "direct" from this sentence.

P11 L3: delete "where" **– DONE.**

P11 L6: This introductory sentence needs to be rewritten for clarity.

- Agreed. Changed to: *"In the start of the cold season, the ice surface was covered by seasonal snow had low firn air-content (e.g. at 7 September 2015, DOY 250 in Fig. 6b). During the cold season the firn air-content increased through time and moving up-glacier, as fresh snow accumulated (e.g. at 23 April 2016, DOY 113 in Fig. 6b)."*

P11 L17-34: Good discussion. It would be worth including in the discussion, the size definition of a target that is considered transparent, compared to those considered good specular or diffuse reflectors at the c-band frequency.

- Good point. We have added a sentence under section 5.2 (third paragraph) to clarify this: *"We consider snow and firn to be Rayleigh scatterers when the particle size is lower than 10% of the wavelength (λ = 5.5 cm) or 0.55 cm, and Mie scatterers up to 10 times the wavelength, or 55 cm (Woodhouse, 2006)."*
  And additionally a sentence to clarify this under the above comment P10 L3-6: *"According to the modified*

*Rayleigh criterion a surface is considered rough in the C-band SAR (using an incidence angle of 38 degrees) when the root-mean-square (rms) surface height variation is more than 1.3 cm, and considered a smooth surface when the rms height variation is less than 0.2 cm (e.g. Lillesand et al., 2004)."*

P11 L21: Re-word: A perennial firn aquifer was found containing liquid water.

- We corrected to: *"Christianson and others (2015) found a perennial firn aquifer containing liquid water in the upper part of the firn area on Holtedahlfonna."*

P11 L22: Clarification required. Is the firn aquifer depth 3.5-15 m, or is it located below the surface 3.5-15 m?

- This has been clarified: *", and they argued that the firn aquifer had a depth below the surface of approximately 3.5 - 15 meters."*

P12 L1: Backscatter values can't be absorbed, please reword/rephrase.

- Agreed. We changed the sentence to*: "It is likely that absorption with limited volume scattering of radar waves in wet snow and firn can give an indication of how deep the Sentinel-1A SAR C-band can penetrate due to a strong sensitivity to wet conditions.".*

P12 L1-6: This contradicts the explanatory statements starting at P11 L28. How can you be certain that SAR penetration depth is increasing? This needs to be reconciled, along with an explanation of possible contributing parallel processes occurring in an evolving snowpack exposed to a penetrating winter cold wave.

- Thank you for pointing this out. When looking at the plots 7 d and e one can see that the backscatter values stay high even when the dry-to-wet transition zone gets deeper. We have tried to better explain what we mean and changed the paragraph to: *"Using the model results we were able to estimate the depths of the intersection between the dry and wet zone in the firn pack (Fig. 7b) and compare this to the backscatter data (Fig. 7c-e). Results indicate increased penetration depths over time and then stabilization once the transition exceeds a certain depth where the radar waves do not reach it anymore. Similar trends between the transient modeled dry-to-wet transition depth and the backscatter time-series for the uppermost part of the glacier was found (Fig. 7c). The deeper the dry-to-wet transition zone, the higher SAR backscatter values (Fig. 7e). This correlation can thus be explained by a mix of volume scattering returns and radar waves sensitivity to wet conditions in snow and firn through time until January 2016."*

P12 L7-14: Suggest reorganization of this section to make it easier for the reader to follow. Also, include a transitional/introductory sentence to link the previous paragraph to this.

Agreed. We have changed to*: "To investigate the penetration depth further, values from black points in Fig. 7a were selected where the backscatter values stabilized in the firn. Values from the same points of dry-to-wet transition depths (Fig. 7b) gave an indication of the depth of the dry-to-wet zone in the backscatter data. The mean transition depth of the point values from Fig. 7b is 1.7 m when all 9 points were included. When only including the upper 6 points with significant correlation > 920 m (showed in Fig.7c and e), a mean transition depth of 2.0 m was found. We speculate that backscatter intensity cannot reflect small*

*changes in snow depth during the dry wintertime due to high volume scattering thus suggesting deeper and temporally constant penetration of the radar waves.*
*In this example, modeled data were used to help interpret time-series of backscatter intensity data. This might be inverted in the future, as SAR backscatter data will be further understood and can potentially be used as refined modeling input. Such information is valuable in remote regions in the high Arctic that are lacking meteorological stations and where it is costly to do field observations."*

P12 L10: Modify. ". . . intensity can not 'be used to' identify. . ."

- Agreed. Use "reflect" in stead of "identify"

P12 L25: Change last word to 'surfaces'**– DONE.**
P13 L23: Explain why an increase in pore space is occurred.

- Changed "pore space" to "air-content".

P13 L24: Delete 'with'**– DONE.**
P13 L28: Case change 'p'**– DONE.**
P13 L29: Rephrase sentence, change 'deeper snow depth'

- Agreed. Changed to: *"In the upper zone of the ablation area and before the rain event more snow was present compared to lower elevations."*

P13 L29-32: This section needs a better explanation of the processes involved to support the conclusions. Why 'higher permittivity'? Could there be another explanation for a sharp increase in backscatter? Snow saturation and superimposed ice is common with these events – are these identifiable?

- Agreed. We have changed to: *"Ice lenses and pipes might thus have been created from this penetration of water in the snow after the rain event, resulting in snow saturation giving stronger permittivity contrast as shown by increasing backscatter values (indicated by yellow color in Fig. 6a. Present around the 400 m elevation line from 17 January to 4 May 2016, DOY 17 to DOY 125)."*

P15 L17: Rearrange wording – 'glaciers with high spatial resolution'

- Agreed. We have changed to: *"Additionally, high spatial resolution Sentinel-1 time-series can be used to measure snow melt parameters on glaciers and with 6-days temporal resolution."*
-
P15 L21: 'SAR-zones' should be 'glacier facies'

- Agreed: Changed to: *"Time-series from 2009-2016 using Radarsat-2 and Sentinel-1A SAR backscatter data, showed relatively stable SAR glacier zones on Kongsvegen and Holtedahlfonna. Dense SAR time-series have a potential for more accurate delineation of glacier facies, compared to using only one acquisition as in previous studies."*

P15 L27-28: Rearrange/rephrase for reader clarity.

- Agreed. We have rephrased to: *"Our findings are important to further understand glaciological processes, and we have shown the potential of combining results from modeled snow/firn evolution with high-resolution SAR backscatter time-series data."*

P17 L5: Rephrase

- Agreed. Changed to: *"After the rain event, but during the same weather situation, the precipitation might have turned into snow on the glacier".*

P29: Minor - Consider switching the color blocks for 'Cold season' and 'Warm season' to reflect customary warm and cool color designations.

- This is a good point, and we have discussed this extensively before initial submission. We have decided to keep the colors like this, because now the red color reflect high backscatter values, and blue colors reflects low backscatter colors. When interpreting SAR backscatter values this representation make more sense in terms of retrieved energy level at the satellite sensor. Wet conditions are now related to blue color (=water), because wetness is strongly affecting the backscatter signal.

**Comments by Anonymous Referee #2:**

4. Are the scientific methods and assumptions valid and clearly outlined? -> Yes, well written and supported by prior literature. My only question is the use of parametric statistics, particularly the Spearman r correlation, for what might be a non-normal and probably not-independent sample (eg. Figure 6c). I believe the authors need to justify this.

- As the reviewer mentions, we cannot assume a normal-distribution (and do not assume a linear relationship between the backscatter and the firn air-content), and therefore avoid using parametric statistics such as the Pearson correlation method. This is the reason why we use the nonparametric Spearman correlation coefficient in e.g. Figure 6c. We have clarified this in the main text: "*To avoid assumptions of a linear relationship between air-content and backscatter, we used the nonparametric Spearman's rank correlation coefficient."* (Page 12, second paragraph). Similarly, we do not assume a linear relationship between the snow/firn wetness (represented as the dry-to-wet transition zone) and backscatter.

6. Is the description of experiments and calculations sufficiently complete and precise to allow their reproduction by fellow scientists (traceability of results)? -> There is a great deal of analysis bound in this work and I have not traced all of the background references and their references. I believe that the literature cited will allow appropriate traceability.

- Experiments and calculations are described and can be traced in literature or in the paper

8. Does the title clearly reflect the contents of the paper? -> The title is short and could benefit from the mention of the five scenarios which is a major contribution.

- We will keep the title as it is. It is short and has a strong message that summarizes the key content of the manuscript.

11. Is the language fluent and precise? -> Some suggested corrections are provided below.

- We agreed with all of the language issues and changed the manuscript accordingly

Page 2, line 21, ". . .1980's have been. . ." **- DONE**
Page 7, line 24, "The TSL migration up-glacier is dependent. . ." **- DONE**
Page 8, line 8, ". . .in the ablation area of, for example, on. . ." **- DONE**
Page 8, line 10, "On addition, in a different example from. . ." (suggestion as 'yet' does not convey anything here) **– DONE, changed to "In addition, a different example from. . ."**
Page 8, line 19: I suggest that "3% lower altitude" does not mean much as a percentage will change with absolute altitude. 3% of x scale or an absolute change of height would provide more clarity. **– Agreed. Changed to***"…show a mean of 21 m lower altitude compared to optical derived TSLs (not taking the outlier into consideration)."***.
Page 10: Section 5.3 is a particularly strong contribution, in my opinion.

- Thank you!

Page 14, line 20 ". . .at the days and time-of-day of SAR. . ." is awkward syntax. There may be a better construction. **– Agreed. Changed to** *"…have been low in some of the SAR acquisitions".*
Page 14, line27, ". . . but be of help. . ." perhaps should be ". . .but can be of help. . ."? **- DONE**
Page 15, line 29, I suggest that the authors mean "Winter rain events" as opposed to "Winter weather events" of which there are many in the Arctic and includes a wide range of precipitation types. **– DONE**

[revised manuscript text omitted]